PREPARED FOR SUBMISSION TO JHEP

# 3d mirror for Argyres-Douglas theories

**Dan Xie**[a,b]

[a] *Yau Mathematics Science center, Tsinghua University, Beijing, 100084, China*
[b] *Department of Mathematics, Tsinghua University, Beijing, 100084, China*

ABSTRACT: 3d mirrors for all 4d $\mathcal{N} = 2$ Argyres-Douglas (AD) theories engineered using 6d $(2,0)$ theory are found. The basic steps are: 1): Find a punctured sphere representation for the AD theories (this is achieved in our previous studies of S duality); 2): Attach a 3d theory for each puncture; 3): Glue together the 3d theory for each puncture. We found the 3d mirror quiver gauge theory for the AD theories engineered using 6d $A$ and $D$ type theories. These 3d mirrors are useful for studying the properties of original 4d theory such as Higgs branch, S-duality, etc; We also construct many new 3d $\mathcal{N} = 4$ SCFTs.

## 1 Introduction

Three dimensional (3d) $\mathcal{N} = 4$ SCFT has very interesting mirror symmetry properties [1], and there are nontrivial maps between physical quantities of two mirror theories $A$ and $B$; For example, the Coulomb branch of theory $A$ is identified with the Higgs branch of theory $B$, and vice versa [1]. This mirror symmetry is quite useful as the physical properties are easier to compute in one theory than the other one.

    The mirror pairs studied in [1] have purely 3d description: theories $A$ and $B$ could arise from the IR limit of purely 3d theories. Another way of getting 3d $\mathcal{N} = 4$ SCFT is to start with a higher dimensional theory with eight supercharges and study its compactification down to 3d. For example, if a four dimensional (4d) $\mathcal{N} = 2$ superconformal field theory (SCFT) is compactified on a circle, one can get three dimensional (3d) $\mathcal{N} = 4$ SCFT [2] in the IR limit, which we call it theory $A$. It would then be interesting to find the 3d mirror theory $B$. If theory $B$ has a simpler description, i.e. it is given by the IR limit of a 3d

---

[1]The moduli space of 3d $\mathcal{N} = 4$ SCFT could have two different branches depending on the transformation properties under the $SU(2)_1 \times SU(2)_2$ R symmetry. The name "Coulomb" branch comes from the fact that the theory could be given as the IR limit of a quiver gauge theory and this branch is identified with the limit of the Coulomb branch of the quiver gauge theory; Similar meaning is applied to the name "Higgs" branch.

quiver gauge theory, then $B$ is very useful to learn interesting properties of theory $A$ (and original 4d theory) which is hard to obtain through other methods. For example, one can find the Higgs branch of the original 4d theory by studying the Coulomb branch of theory $B$.

In general, there is no systematic way to find 3d mirror pairs. Type IIB Brane construction is powerful in finding mirror for 3d linear (or cyclic) quivers [3–6]. The mirror theory for 3d abelian gauge theories was studied in [7]. One can also find mirror pairs by studying circle compactification of 4d $\mathcal{N} = 2$ class $\mathcal{S}$ theories, where theory $B$ is given by the IR limit of star-shaped quiver [8]. The mirror for circle compactification of some 4d Argyes-Douglas (AD) theories are found in [9, 10] (see [11] for related mathematical study). These 3d mirrors are important tools to study properties of 4d theory, i.e. the Higgs branch of 4d theory [12] and the $S$ duality property [13].

The 3d mirrors of **most** AD theories found in [10, 14, 15] are not found [2]. The purpose of this paper is to fill this gap and find the 3d mirror for **all** the AD theories constructed from 6d $(2, 0)$ theories. The punctured Riemann surface construction [21] for class $\mathcal{S}$ theory is very useful to find its 3d mirror [8]: one associates a quiver tail for each puncture and the full mirror quiver is derived by gluing these quiver tails. The most important discovery of this paper is that similar strategy works for all the other AD theories, and the crucial ingredient is the punctured sphere representation used in studying S duality [22, 23]. Given the punctured sphere representation of AD theories, one associates a quiver tail for each puncture and the mirror quiver $B$ is constructed by gluing these quiver tails! We summarize the detailed strategy of finding 3d mirror for AD theories [3]:

1. Find a punctured Riemann sphere representation of AD theory [22, 23]. There are typically three kinds of punctures: black, blue, and red. Each puncture has a label, i.e. a Young Tableaux for $A$ type, see section 2 for more details. See figure. 1 for an example [4].

2. Attach a quiver tail for each puncture. The quiver tail of red and black puncture is quite similar to that of [8], which is often a linear quiver. The quiver tail for the black puncture is more subtle: the adjoint matters are needed. See figure. 1.

3. Glue above quiver tails together. This is the most difficult part of the construction, and the rule is found by using various predictions from 3d mirror: the match of Higgs (Coulomb) branch of theory $A$ and Coulomb (Higgs) branch of theory $B$, etc. See the rule listed in figure. 2.

Several examples are listed in the figure. 3. The above strategy is similar to what is used in finding 3d mirror for class $\mathcal{S}$ theories [8], and the stories presented here is much more general and the case [8] can be regarded as a special case. If the AD theories are constructed using 6d $(2, 0)$ theory of $A$ and $D$ type, the 3d mirror has a quiver gauge theory

---

[2]The author proposed 3d mirror for $(A_1, A_2)$ theory in [9], some other examples are found in [12], and see [16] for discussion of the 3d mirror of $(A_1, A_{2N})$ and $(A_1, D_{2N+1})$ theories. See also [17–20] for 3d mirror of some AD theories.

[3]When we talk about 3d mirror for AD theories, we always mean the corresponding 3d $\mathcal{N} = 4$ SCFT derived by compactifying 4d AD theory on a circle.

[4]Notice that this is not the punctured sphere where one uses to engineer AD theory from 6d $(2, 0)$ SCFT.

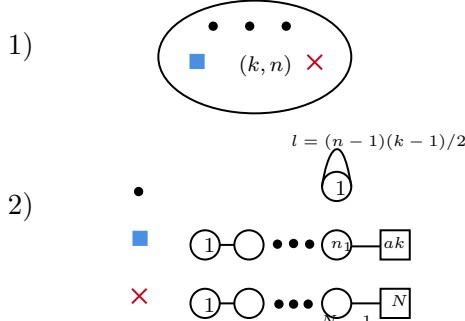

**Figure 1**. 1): A 4d Argyres-Douglas theory engineered using 6d $A_{N-1}$ $(2,0)$ theories can be represented by a punctured sphere, and there are three types of punctures: blue, black, and red; Here the black puncture is taken to be the simplest one (with Young Tableaux [1]), and blue and red puncture are taken to be the maximal ones: the blue one has flavor symmetry $U(n_1)$ and the red one has $SU(N)$ flavor symmetry. There is also an extra label $(k, n)$ on the punctured sphere. 2): The quiver tail for each type of puncture; and there is adjoint matter for black puncture. The numeric numbers are related as $N = na + n_1$, with $a$ the number of simple black punctures, so the theory is specified by four numbers $(a, n_1, n, k)$.

description, and more generally the 3d mirror is constructed by gauging strongly coupled 3d SCFTs. It is quite satisfactory that there is a simple and uniform way of finding 3d mirror for all 4d $\mathcal{N} = 2$ theories constructed using 6d $(2,0)$ theories.

The 3d mirror of AD theory is quite useful in studying the properties of 4d AD theory: a): it confirms the S duality conjectures of these theories [22, 23]: the S duality could be found by decomposing the 3d mirror into various pieces corresponding to the mirror of AD matter, see [13] for how to use 3d mirror to find S duality of AD theory; b): it predicts the Higgs branch of 4d theories: the Coulomb branch of the mirror quiver can be computed and they agree with the Higgs branch result of AD theories found in [24]. It also has interesting applications for the study of 3d $\mathcal{N} = 4$ SCFTs: we find a large class of new interesting theories and some of them can be useful to construct new Chern-Simons matter theory.

This paper is organized as follows: section 2 reviews the 4d AD theory constructed in [10, 14, 15]; section 3 discusses the 3d mirror for general AD theories; section 4 discusses some applications of 3d mirror of AD theories; finally, a conclusion is given in section 5.

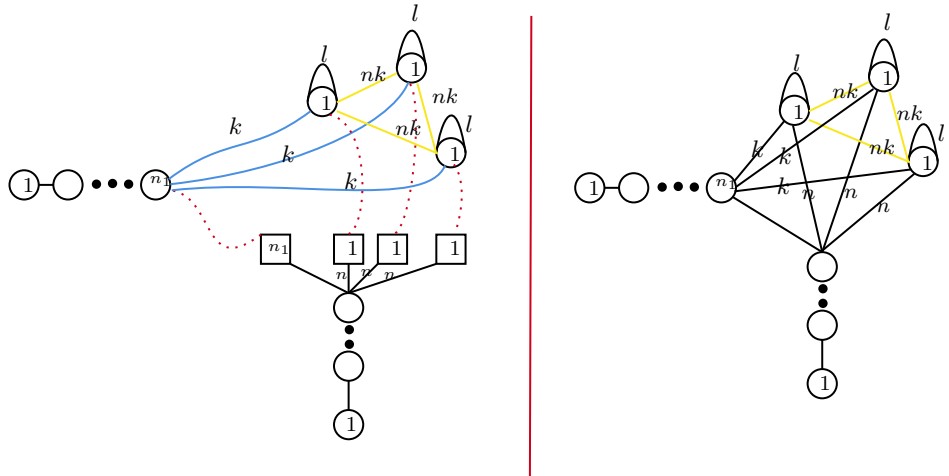

**Figure 2**. Left: the gluing rule for the quiver tails listed in figure. 1: a) there are $nk$ edges between black tails; b): there are $k$ edges between a blue tail and a black tail; c): We spray the flavor quiver node for the red tail so the flavor symmetry is $U(n_1) \times U(1)^a$, and there are $n$ edges for every $U(1)$ flavor node; the $U(n_1)$ flavor node is glued with the blue tail, and $U(1)$s are glued with black tail. Right: the final mirror quiver. $l$ is given by $\frac{(n-1)(k-1)}{2}$.

$(A_{n-1}, A_{k-1})$ theory     $l = (n-1)(k-1)/2$

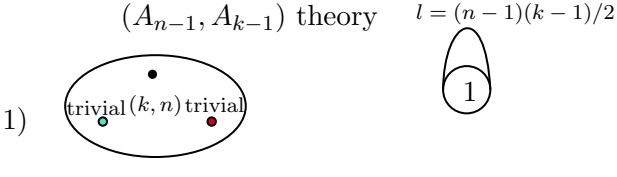

1)

$D_{n+k}(SU(n))$ theory

$l = (n-1)(k-1)/2$

2)

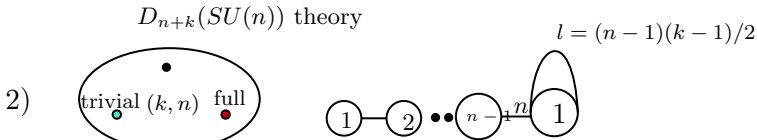

**Figure 3**. 1): The 3d mirror for $(A_{n-1}, A_{k-1})$ theory, here $n, k$ is coprime $((n,k) = 1)$. 2): The 3d mirror for $D_{n+k}(SU(n))$ theory with $(n,k) = 1$ and $k > 0$, the 3d mirror for the case $k < 0$ is given in section 3. In both examples, the black puncture is the simplest one with label $[1]$.

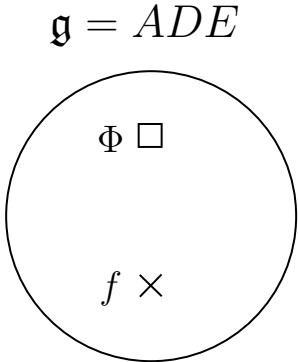

**Figure 4**. A 4d Argyres-Douglas theory is constructed by putting a 6d $(2,0)$ theory of type $\mathfrak{g}$ on a sphere with one irregular singularity and one regular singularity. The irregular singularity is labeled by $\Phi$, see 2.1, and the regular singularity is labeled by $f$.

## 2  AD theories from 6d $(2,0)$ SCFT

### 2.1  Basic construction

We give a review for the construction of AD theory from 6d $(2,0)$ theory, see [25] (which I take some parts from) for more details. One can engineer a large class of 4d $\mathcal{N}=2$ SCFTs by putting a 6d $(2,0)$ theory of type $\mathfrak{j}=ADE$ on a sphere with an irregular singularity and a regular singularity [10, 14, 15, 21, 26], see figure. 4. The Coulomb branch is captured by a Hitchin system with singular boundary conditions near the singularity. The Higgs field of the Hitchin system near the irregular singularity takes the following form,

$$\Phi = \frac{T}{z^{2+\frac{k}{b}}} + \dots. \tag{2.1}$$

Here $T$ is determined by a positive principle grading of Lie algebra $\mathfrak{j}$ [27], and is a regular semi-simple element of $\mathfrak{j}$. $k > -b$ and is an integer. Subsequent terms of the Higgs field are chosen such that they are compatible with the leading order term (essentially the grading determines the choice of these terms). We call them $J^{(b)}[k]$ type irregular puncture. Theories constructed using only above irregular singularity can also be engineered using a three dimensional singularity in type IIB string theory as summarized in table 1[28]. One can add another regular singularity which is labeled by a nilpotent orbit $f$ of $\mathfrak{j}$ (We use Nahm labels such that the trivial orbit corresponding to regular puncture with maximal flavor symmetry). A detailed discussion about these defects can be found in [29]. So a theory is specified by four labels $<\mathfrak{j}, b, k, f>$, here $\mathfrak{j}$ denotes type of 6d $(2,0)$ SCFT, $b, k$ denotes irregular singularity, and $f$ denotes regular singularity.

To get non-simply laced flavor groups, we need to consider the outer-automorphism twist of ADE Lie algebra and its Langlands dual. A systematic study of these AD theories was performed in [15]. Denoting the twisted Lie algebra of $\mathfrak{j}$ as $\mathfrak{g}^\vee$ and its Langlands dual as $\mathfrak{g}$, outer-automorphisms and twisted algebras of $\mathfrak{j}$ are summarized in table 2. The irregular singularity of regular semi-simple type is also classified as in table 3 with the following

| j | b | Singularity |
|---|---|---|
| $A_{N-1}$ | $N$ | $x_1^2 + x_2^2 + x_3^N + z^k = 0$ |
| | $N-1$ | $x_1^2 + x_2^2 + x_3^N + x_3 z^k = 0$ |
| $D_N$ | $2N-2$ | $x_1^2 + x_2^{N-1} + x_2 x_3^2 + z^k = 0$ |
| | $N$ | $x_1^2 + x_2^{N-1} + x_2 x_3^2 + z^k x_3 = 0$ |
| $E_6$ | $12$ | $x_1^2 + x_2^3 + x_3^4 + z^k = 0$ |
| | $9$ | $x_1^2 + x_2^3 + x_3^4 + z^k x_3 = 0$ |
| | $8$ | $x_1^2 + x_2^3 + x_3^4 + z^k x_2 = 0$ |
| $E_7$ | $18$ | $x_1^2 + x_2^3 + x_2 x_3^3 + z^k = 0$ |
| | $14$ | $x_1^2 + x_2^3 + x_2 x_3^3 + z^k x_3 = 0$ |
| $E_8$ | $30$ | $x_1^2 + x_2^3 + x_3^5 + z^k = 0$ |
| | $24$ | $x_1^2 + x_2^3 + x_3^5 + z^k x_3 = 0$ |
| | $20$ | $x_1^2 + x_2^3 + x_3^5 + z^k x_2 = 0$ |

**Table 1**. Three-fold isolated quasi-homogenous singularities of cDV type corresponding to the $J^{(b)}[k]$ irregular punctures of the regular-semisimple type in [14]. These 3d singularity is very useful in extracting the Coulomb branch spectrum, see [28].

| $j$ | $A_{2N}$ | $A_{2N-1}$ | $D_{N+1}$ | $E_6$ | $D_4$ |
|---|---|---|---|---|---|
| Outer-automorphism $o$ | $Z_2$ | $Z_2$ | $Z_2$ | $Z_2$ | $Z_3$ |
| Invariant subalgebra $\mathfrak{g}^\vee$ | $B_N$ | $C_N$ | $B_N$ | $F_4$ | $G_2$ |
| Flavor symmetry $\mathfrak{g}$ | $C_N^{(1)}$ | $B_N$ | $C_N^{(2)}$ | $F_4$ | $G_2$ |

**Table 2**. Outer-automorphisms of simple Lie algebras $j$, its invariant subalgebra $g^\vee$ and flavor symmetry $g$ from the Langlands dual of $g^\vee$.

form,

$$\Phi = \frac{T^t}{z^{2+\frac{k_t}{b_t}}} + \dots \tag{2.2}$$

Here $T^t$ is an element of Lie algebra $\mathfrak{g}^\vee$ or other parts of the decomposition of $j$ under outer automorphism. $k_t > -b_t$, and the novel thing is that $k_t$ take half-integer value or in thirds ($\mathfrak{g} = G_2$). We could again add a twisted regular puncture labeled by a nilpotent orbit $f$ of $\mathfrak{g}$. A theory is then labeled by following data $< j, o, b_t, k_t, f >$. Here $j$ is the type of 6d $(2,0)$ SCFT, $o$ is the outer automorphism twist we use, $b_t, k_t$ denotes the irregular singularity, and $f$ denotes regular singularity.

## 2.2 Two generalizations and geometric representation

It was found in [22] that one can find more SCFTs by using other kinds of irregular singularities which are different from those listed in last subsection. This class of theories are necessary for studying S dualities of these theories. The generalizations are the following:

| $j$ with twist | $b_t$ | SW geometry at SCFT point | $\Delta[z]$ |
|---|---|---|---|
| $A_{2N}/Z_2$ | $2N+1$ | $x_1^2 + x_2^2 + x^{2N+1} + z^{k+\frac{1}{2}} = 0$ | $\frac{4N+2}{4N+2k+3}$ |
| | $2N$ | $x_1^2 + x_2^2 + x^{2N+1} + xz^k = 0$ | $\frac{2N}{k+2N}$ |
| $A_{2N-1}/Z_2$ | $2N-1$ | $x_1^2 + x_2^2 + x^{2N} + xz^{k+\frac{1}{2}} = 0$ | $\frac{4N-2}{4N+2k-1}$ |
| | $2N$ | $x_1^2 + x_2^2 + x^{2N} + z^k = 0$ | $\frac{2N}{2N+k}$ |
| $D_{N+1}/Z_2$ | $N+1$ | $x_1^2 + x_2^N + x_2x_3^2 + x_3z^{k+\frac{1}{2}} = 0$ | $\frac{2N+2}{2k+2N+3}$ |
| | $2N$ | $x_1^2 + x_2^N + x_2x_3^2 + z^k = 0$ | $\frac{2N}{k+2N}$ |
| $D_4/Z_3$ | $4$ | $x_1^2 + x_2^3 + x_2x_3^2 + x_3z^{k\pm\frac{1}{3}} = 0$ | $\frac{12}{12+3k\pm1}$ |
| | $6$ | $x_1^2 + x_2^3 + x_2x_3^2 + z^k = 0$ | $\frac{6}{6+k}$ |
| $E_6/Z_2$ | $9$ | $x_1^2 + x_2^3 + x_3^4 + x_3z^{k+\frac{1}{2}} = 0$ | $\frac{18}{18+2k+1}$ |
| | $12$ | $x_1^2 + x_2^3 + x_3^4 + z^k = 0$ | $\frac{12}{12+k}$ |
| | $8$ | $x_1^2 + x_2^3 + x_3^4 + x_2z^k = 0$ | $\frac{8}{12+k}$ |

**Table 3**. SW geometry of twisted theories at the SCFT point. Here we also list the scaling dimension of coordinate $z$. All $k$'s in this table are integer valued and the power of $z$ coordinate in singularity is equal to $k_t$ used in equation 2.2.

1. The irregular singularity takes the following block diagonal form

$$\Phi = \begin{bmatrix} \Phi_1 & 0 \\ 0 & \Phi_2 \end{bmatrix} = \begin{bmatrix} \frac{T_1}{z} & 0 \\ 0 & \frac{T_2}{z^{2+\frac{k}{n}}} \end{bmatrix} + \dots \tag{2.3}$$

Here $\Phi_1$ has regular singularity, and $\Phi_2$ is an irregular singularity listed in last subsection. This type of singularity can be regarded as the combination of regular and irregular singularity.

2. For the irregular singularity listed in the last subsection, if $k$ and $b$ are not co-prime, it is possible to consider the degenerating case, i.e. some of the coefficients of the higher order pole is set to be the same. For example, let's consider $\mathsf{j} = A_3, k = 6, b = 4$. The generic coefficients of Higgs field considered in last section take the from

$$\Phi = \begin{pmatrix} a_1 \begin{pmatrix} 1 & 0 \\ 0 & -1 \end{pmatrix} & 0 \\ 0 & a_2 \begin{pmatrix} 1 & 0 \\ 0 & -1 \end{pmatrix} \end{pmatrix} \frac{1}{z^{1+\frac{6}{4}}} + \dots \tag{2.4}$$

The leading order term has two different coefficients $a_1$ and $a_2$. For the degenerating case, the leading order coefficients take the following form instead:

$$\Phi = \begin{pmatrix} a \begin{pmatrix} 1 & 0 \\ 0 & -1 \end{pmatrix} & 0 \\ 0 & a \begin{pmatrix} 1 & 0 \\ 0 & -1 \end{pmatrix} \end{pmatrix} \frac{1}{z^{1+\frac{6}{4}}} + \dots \tag{2.5}$$

and here the leading order term has only one coefficient. If we'd like to get a conformal field theory, it was shown in [22] that the irregular part should have the same pattern of degeneracy. The first order part, on the other hand, could take arbitrary degenerating form.

Given the generalizations of irregular singularities, it is possible to represent our theory by an auxiliary punctured Riemann sphere (notice that this is an extra sphere), see figure. 5. Here we take $j = A_{N-1}$ type as example, other cases are similar. The original sphere in the $(2,0)$ construction involves an irregular singularity and a regular singularity. The basic idea of finding an auxiliary sphere is to represent a single irregular singularity by several punctures of an extra sphere. We use a red puncture to denote the regular singularity, a blue puncture for the regular part inside the irregular singularity 3.6, and several black punctures for the irregular blocks in irregular singularity. Each puncture has a Young Tableaux with different size. The rule for assigning a Young Tableaux to a red and blue puncture is the same as that found in [21], as they are both just regular singularities. For the irregular singularity, they take the following general form

$$\Phi = \frac{1}{z^{2+\frac{np}{nq}}} diag(\underbrace{a_1 j_q, \ldots, a_1 j_q}_{n_1}, \ldots, \ldots, \underbrace{a_s j_q, \ldots, a_s j_q}_{n_s}) + \ldots \qquad (2.6)$$

Here $j_q$ is a standard diagonal matrix depending on integer $q$, and the detailed form is not important here (see [10]). We represent this irregular singularity by $s$ black punctures, and each puncture has a size $n_1$. The detailed form of the Young Tableaux at each puncture is determined by the form of first order coefficients!

The above representation is similar to the Class $\mathcal{S}$ theory where there are only one type of puncture (red puncture) [21]. Here we need three kinds of punctures, and there are some further important differences:

1. There is only one red and one blue puncture (both of them could be **trivial**), and arbitrary number of black puncture (there should be at least one nontrivial black puncture).

2. One need an extra pair of co-prime integers $(p, q)$ to indicate the theory type, this pair gives the slope of the irregular part of the irregular singularity ($\frac{k}{b} = \frac{p}{q}$), see 2.1.

3. When $(p, q) = (k, 1)$, $k > 0$, the blue puncture and black puncture are of the same type. When $(p, q) = (0, 1)$, all three punctures are of the same type, this actually represents class $S$ theory engineered using only regular punctures on sphere [21].

4. The size of Young tableaux is related as follows

$$Y_{red} = Y_{blue} + q \sum_{black} Y_i \qquad (2.7)$$

The basic theory (called AD matter in [22]) is represented by a sphere with three punctures: one red, one blue and one black puncture.

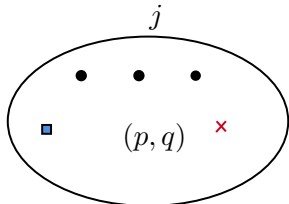

**Figure 5**. A configuration shown in figure. 4 is now represented by a different punctured sphere: here the red puncture represents the regular singularity, the blue puncture represents the regular part in irregular singularity, and the black punctures represent the irregular blocks inside irregular singularity.

The weakly coupled gauge theory description of general AD theory is found by degenerating above punctured Riemann surface into three punctured spheres [22, 23]: the rule is that the blue puncture of one three sphere is connected with the red puncture of the other sphere, and each degenerating three sphere should be the allowed type: in the general case, it should have one blue, one red and one black puncture.

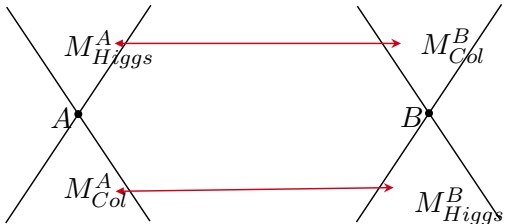

**Figure 6**. The basic feature of 3d mirror symmetry: the Higgs branch of theory $A$ is identified with the Coulomb branch of theory $B$, and vice versa.

## 3 3d mirror for AD theories

Let's now put 4d $\mathcal{N} = 2$ theory discussed in last section on a circle with finite radius. The resulting low energy theory on the Coulomb branch can be described by the moduli space of the Hitchin system $\mathcal{M}_{Hit}$ [30]. The Hitchin moduli space has a Hitchin map $\pi : \mathcal{M}_{Hit} \to B$ [31], where the generic fibre of this map is an open set inside an abelian variety (the abelian variety has half dimension of the Hitchin moduli space) [31]. The Coulomb branch of both 4d and 3d theory can be described by Hitchin fiberation. For four dimensional theory, only the base $B$ describes the Coulomb branch moduli, and the low energy photon coupling is given by the complex structure of the abelian variety. For three dimensional theory, the fibre is also regarded as the vacua moduli (the reason is that in three dimension, one can perform duality on abelian gauge fields to get scalar fields which parameterize the fibre). The Higgs branch of 3d theory does not receive quantum corrections and is the same as the Higgs branch of the parent 4d theory.

We are interested in the most singular point of $M_{Hit}$ which should be a three dimensional $\mathcal{N} = 4$ SCFT, and we call it theory $A$. One might get the properties of this 3d SCFT by taking the radius of the compactified circle to be zero, and flow to the deep IR. Conjecturally, the Coulomb branch of this 3d SCFT $A$ is given by the dense open set $\mathcal{M}^*_{Hit}$ of $\mathcal{M}_{Hit}$. We would like to find a 3d SCFT $B$ which is a mirror theory of theory $A$. In some cases, the mirror SCFT can be described as the IR SCFT of a quiver gauge theory (We want to emphasize that this is not always possible); If this is the case, the Higgs branch of theory $B$ has a classical description as the hyperkahler quotient, which is often called Nakajima quiver variety [32]. In this case we have following useful map:

$$\mathcal{M}^*_{Hit} = quiver\ variety \tag{3.1}$$

Mathematically, some examples of above equivalence were shown in [33]. Physically, the results in [33] is interpreted as 3d mirror symmetry [8, 10].

Let's recall some basic maps of 3d mirror symmetry [1]:

1. The Coulomb branch of theory $A$ is identified with Higgs branch of theory $B$, and vice verse. The simple checks are: the dimension should match, and the flavor symmetries should also match.

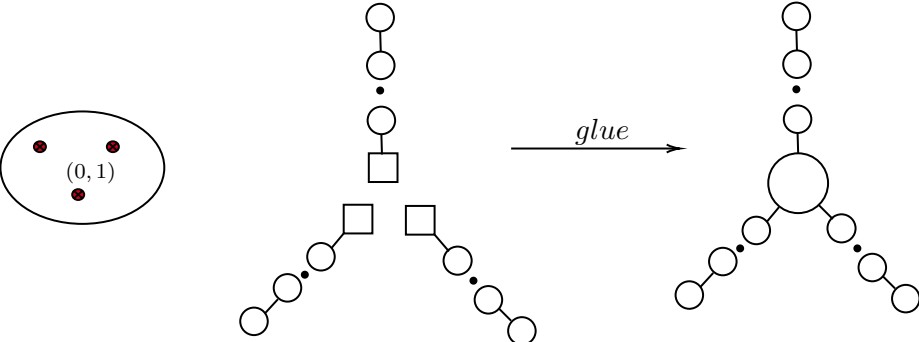

**Figure 7**. Left: a 4d $\mathcal{N} = 2$ theory is engineered by putting $A_{N-1}$ $(2,0)$ theory on a three punctured sphere. Middle: We associate a linear quiver tail for each red puncture, whose form is given in 3.2, and the flavor node of the quiver tail is $U(N)$. Right: the quiver tails are glued together by gauging diagonally the flavor nodes of the quiver tails, and this gives the 3d mirror for the left theory.

2. The mass deformation which would deform the Coulomb branch (usually lift the Higgs branch), mapped to the FI parameters on the Higgs branch of the mirror theory.

The map is shown schematically in figure. 6. Notice that it is possible that the theory $A$ has just one type of moduli space, and it is still possible to find its mirror theory. An example is the IR SCFT of $U(1)$ gauge theory coupled with a single hypermultiplet, which has only Coulomb branch but no Higgs branch; its mirror theory is just a free hypermultiplet, which has only a Higgs branch but no Coulomb branch.

The 3d mirror for class $\mathcal{S}$ theory is successfully found in [8] (This corresponds to $(0,1)$ class of theories discussed in last section). The idea is the following (here we take $\mathfrak{j} = A_{N-1}$ as examples). Recall that the geometric representation for this type of theories has only regular singularities and so there are only red punctures (see 2.2); and the 3d mirror is found as follows:

- For each puncture with label $Y$, one associates a linear quiver tail whose structure is determined by $Y$. There is a $U(N)$ flavor node at the end. For example, if the red puncture has the label $[h_1, h_2, \ldots, h_s]$, then the quiver tail would be

$$\boxed{U(N)} - U(r_1) - U(r_2) - \ldots - U(r_{s-1}) \tag{3.2}$$

with $r_i = \sum_{j=i+1}^{j=s} h_j$.

- The mirror is formed by gauging the diagonal $U(N)$ of the flavor quiver nodes of these quiver tails, and we get a star-shaped quiver.

See figure. 7 for an example.

The 3d mirror for $(k, 1)$ class of theories is found in [10]. The 3d mirror is found as follows: first attaching a complicated quiver for the irregular singularity, and a quiver tail for the regular singularity; second, the 3d mirror is found by gluing the above two quivers

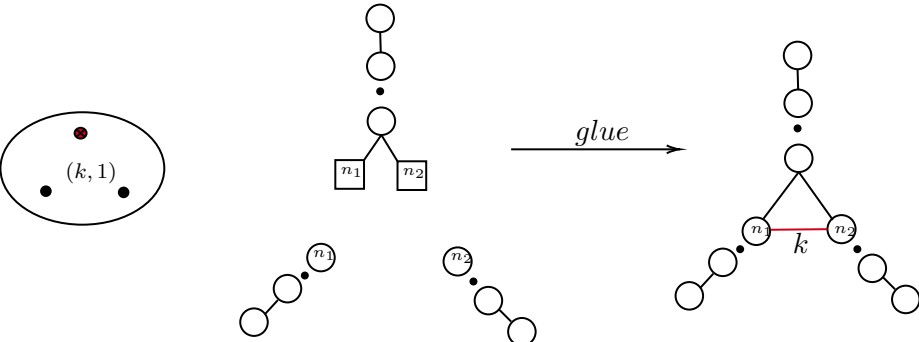

**Figure 8**. Left: a three punctured sphere of $(k, 1)$ type theories. Middle: Attach a quiver tail for each puncture. There is a flavor $U(N)$ node for the quiver tail of red puncture and we decompose it into $U(n_1) \times U(n_2)$; notice that the flavor node $U(n_i)$ for the quiver tail of the black puncture is gauged. Right: the gluing rule is following: a): there are $k$ edges between the end nodes of the black punctures; b): the $U(N)$ flavor node of the red puncture is sprayed as $U(n_1) \times U(n_2)$, and we identify the flavor nodes with the end nodes of black punctures.

together. The 6d construction shown in figure. 4 is very useful in extracting the 3d mirror. One might try to generalize the above construction to other AD theories: assigning a quiver for the general irregular singularity and regular singularity, and then glue the quiver together. However, we find it quite difficult to do so as there is no simple rule for assigning a quiver for an arbitrary irregular singularity. To overcome the difficulty of assigning a quiver for general irregular singularity, we found that re-interpreting the result in [10] by using the geometric representation of figure. 5 is very illumilating. This class of theory (type $(k, 1)$) is represented by a punctured sphere with two kinds of punctures: red one and black one. We reinterpret the rule of finding its 3d mirror as follows:

- One has a quiver tail for a red puncture, and the rule is the same as 3.2. One has a quiver tail for black puncture according to its Young Tableaux. The rule is also the same as that listed in 3.2, and the **difference** is that the flavor quiver node in 3.2 is also gauged: for instance, if a black puncture has label [1], the associated quiver is a gauged $U(1)$ node.

- The gluing rule is following: there are $k$ edges between the end nodes of quiver tails of black puncture. For the red puncture, we choose following subgroup of its flavor quiver node $\prod_i U(n_i)$ where $n_i$ is the size of Young Tableaux of $i$th black puncture; Here we have $\sum n_i = N$. and the gluing is simply identifying these flavor nodes with the end quiver nodes of black punctures.

- The gluing rule is consistent due to the constraint shown in equation 2.7 [5].

An example is shown in figure. 8.

---

[5]Since there is no blue puncture and $q = 1$ in formula 2.7, we have $N = \sum n_i$, here $N$ is the size of the Young Tableaux of red puncture, and $n_i$ is the size of the Young Tableaux of the $i$th black puncture.

The construction of 3d mirror of above two class of theories are quite suggestive, since it shows the following pattern:

1. Find the quiver tails for the red, blue, and black punctures.

2. The 3d mirror is formed by finding appropriate gluing rules, which would glue together the quiver tails of each puncture.

Once we realize the above patterns, it is quite natural to try to implement above strategy for other type of AD theories. Very amusingly, the above ideas work perfectly for all the AD theories constructed using 6d $(2,0)$ theories.

## 3.1 A type theories

Let's now try to find the 3d mirror for general $A$ type AD theory. These theories are represented by a punctured sphere with three kinds of punctures: red, black and blue. Each puncture has a Young Tableaux and there is also an extra label $(k, n)$ for this class of theories.

To find its 3d mirror, we need to find the quiver tail for each puncture, and then find a gluing rule. The quiver tail for the red puncture is the one given in formula 3.2. The quiver tail for the blue puncture is the same as the tail for the red puncture with only one difference: the end node is gauged. The extra gauging can be understood as follows: the flavor symmetry for the blue puncture is actually $U(m)$ while that of red puncture is $SU(N)$. Since the number of $U(1)$ factors (this gives the number of FI parameter) of the mirror should be the same the number of mass parameters of the original theory, the quiver tail for the blue puncture should have an extra gauged $U(1)$, which explains the fact that the end node of blue puncture should be gauged. To match with Coulomb branch contribution of the blue puncture, we also need to add extra hypermultiplets on the $U(m)$ node, and the number is given by

$$k \sum n_i \tag{3.3}$$

Here $n_i$ is the size of the Young Tableaux of the $i$th black puncture.

So what is the quiver tail for the black puncture? To answer this question, we consider the simplest AD theory which is represented by a sphere with just one black puncture whose Young Tableaux is $[1]$ (the blue and red puncture are trivial), and the type of the theory is labeled by $(k, n)$. This theory is engineered by a six dimensional $A_{n-1}$ $(2,0)$ theory on a sphere with only an irregular singularity of the type:

$$\Phi = \frac{T}{z^{2+\frac{k}{n}}} + \dots \tag{3.4}$$

Here $T$ is regular semi-simple, and here we assume $n \geq 0$, and $k, n$ are co-prime. This class of theory is also called $(A_{n-1}, A_{k-1})$ theory [34]. This theory has no Higgs branch, and the Coulomb branch dimension is equal to $\frac{(k-1)(n-1)}{2}$ [10]. If there is indeed a mirror quiver for this theory, it must have just a single $U(1)$ quiver node [6] . Now to match the Coulomb

---

[6]If there is no fundamental matter, the overall $U(1)$ of the quiver gauge theory is decoupled, so if the quiver has only one quiver node, its Coulomb branch is empty.

branch dimension of the original theory, we must add extra $l = \frac{(k-1)(n-1)}{2}$ adjoint on the $U(1)$ quiver node, see figure. 9. We now propose that the quiver shown in figure. 9 is indeed the 3d mirror for the black puncture with Young Tableaux [1]. A simple check is that it gives the correct answer for $n = 1$. More checks would be given later.

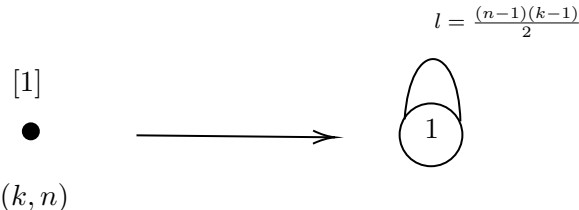

**Figure 9**. The quiver tail for a black puncture with label [1]. The punctured sphere has the label $(k, n)$.

For the general black puncture, we can form a quiver tail (the end node is gauged) according to its Young Tableaux, and the only difference is to add an extra $l = \frac{(n-1)(k-1)}{2}$ adjoints on the end node. See figure. 10. With this proposal, we now have the quiver tails for all kinds of punctures, see figure. 10.

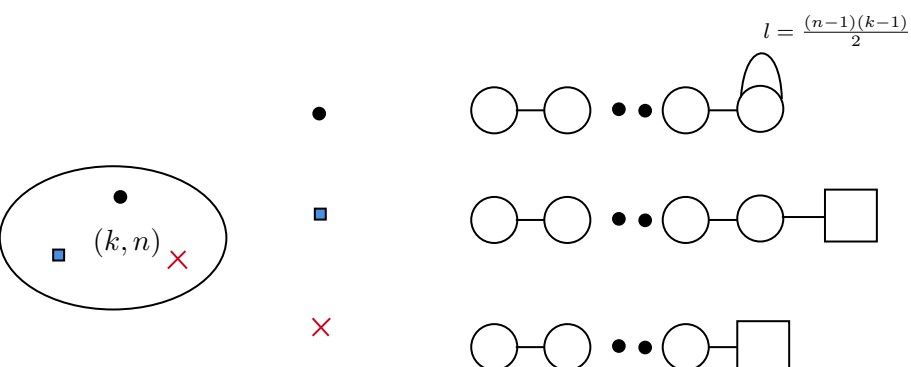

**Figure 10**. Quiver tails for three kinds of punctures of type $(k, n)$ theories. The flavor node of the red tail is $N$, and the flavor tail of the blue tail is $k \sum n_i$ with $n_i$ the size of Young Tableaux of the $i$th black puncture. We also have the relation $N = m + n \sum n_i$.

The next question is the gluing rule. Let's denote the rank of the end node of the red tail as $N$, that of the blue tail as $m$, and the ranks of the black tails as $n_1, \ldots, n_s$. There is a constraint on these numbers (see formula 2.7):

$$N = m + n \sum_{i=1}^{s} n_s \tag{3.5}$$

Now let's state the gluing rule:

- The Higgs branch dimension of the mirror should be equal to the Coulomb branch of the original theory, which is computed in [10]. By computing some examples, we

find following rules: 1) there are $nk$ edges between end nodes of black tails; 2) there are $k$ edges between end nodes of blue and black nodes. See figure. 11.

- We decompose the flavor node of the red puncture as $U(N) = U(m) \times U(n_1) \times \ldots \times U(n_s)$, and the number of edges for black flavor symmetries are $n$. The gluing rule is to simply gauge these nodes with the end nodes of black and blue tails.

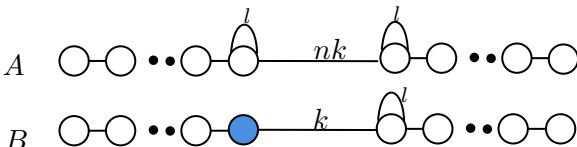

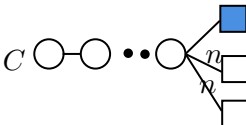

**Figure 11**. The gluing rule for $(k, n)$ type theory: a): There are $nk$ edges between black quiver tails; b): There are $k$ edges between black and blue tails. c): The flavor node of the red tail is decomposed as $U(m) \times U(n_1) \times \ldots \times U(n_s)$, and the multiplicity of the black flavor node is $n$.

Using above rules ,we can find 3d mirror for several interesting class of theories, see figure. 12.

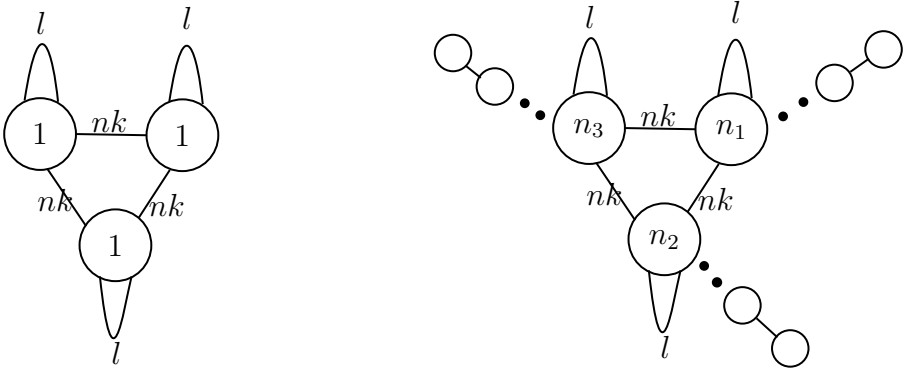

**Figure 12**. Here $l = \frac{(n-1)(k-1)}{2}$. Left: The 4d theory is represented by a sphere with three black punctures with type [1], and one trivial red puncture, and one trivial blue puncture. This theory is actually $(A_{3n-1}, A_{3k-1})$ theory, and the 3d mirror was found here. Right: The original theory is represented by a sphere with three generic black punctures of type $(k, n)$, and here the 3d mirror is given.

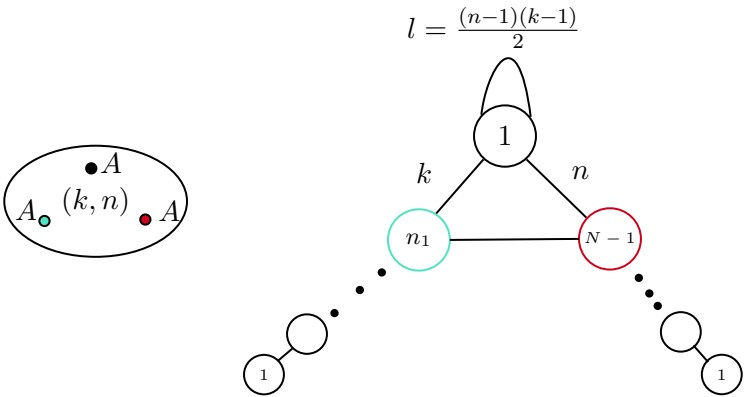

**Figure 13**. This theory has flavor symmetry $SU(N) \times U(n_1)$, and is represented by a sphere with one maximal blue puncture ($Y = [1^{n_1}]$), one maximal red puncture ($Y = [1^N]$), and one simple black puncture ($Y = [1]$). The 3d mirror is shown on the right. Here $N = n_1 + n$.

As an important example, we found the 3d mirror for AD matter found in [22]. This class of theory is engineered using the irregular singularity (here $\mathfrak{g} = A_{N-1}$):

$$\Phi = \begin{bmatrix} \Phi_1 & 0 \\ 0 & \Phi_2 \end{bmatrix} = \begin{bmatrix} \frac{T_1}{z} & 0 \\ 0 & \frac{T_2}{z^{2+\frac{k}{n}}} \end{bmatrix} + \dots \tag{3.6}$$

Here $T_1$ has size $n_1$, and there is a maximal regular singularity $f$, see figure. 4. This theory has flavor symmetry $U(n_1) \times SU(N)$. It is represented by a sphere with one maximal blue puncture, one maximal red puncture, and one black puncture of type [1]. This is a type $(k, n)$ theory. According to our proposal, its 3d mirror is found in figure. 13. Let's now make several checks for the mirror quiver shown in figure. 13:

1. The Coulomb branch dimension of original 4d theory is (see page 28 of [35]):

$$n_C = \frac{(n + k - 1)(2n_1 + n + 1)}{4} \tag{3.7}$$

   This equals to the Higgs branch dimension of the quiver shown in figure. 13. We assume the mirror quiver has a pure Higgs branch, and its dimension is given by the the difference of hypermultiplets and vectormultiplets. There is no flavor symmetry on the Coulomb branch of original theory, and so there is no flavor symmetry on the Higgs branch of the mirror theory. This implies that the mirror quiver should have no flavor quiver node, which is the case for the quiver shown in figure. 13.

2. The Higgs branch of 4d AD matter theory is in general a mixed branch, i.e. the generic point of the Higgs branch consists of free hypermultiplets and an interacting AD theory which has no Higgs branch [24]. The Higgs branch part are described by following variety:

$$\overline{\mathcal{O}_q} \cap S_f; \tag{3.8}$$

Here $\mathcal{O}_q$ is the nilpotent orbit with partition $[\underbrace{n+k,\ldots,n+k}_{n_1},n]$, and $f = [(n+k-1)^{n_1}, 1^{n+n_1}]$. The dimension of Higgs branch is simply the difference of the dimension of $\overline{\mathcal{O}_q}$ and $S_f$, and direct computation gives us that

$$n_H = n_1^2 + nn_1 + \frac{n^2}{2} - \frac{n}{2}. \tag{3.9}$$

The interacting theory part is given by $(A_{n-1}, A_{k-1})$ theory, and this theory has only a Coulomb branch with dimension $\frac{(n-1)(k-1)}{2}$.

We now show that the mirror theory of the quiver shown in figure. 13 recovers the mixed branch structure found in [24]. Firstly, one can check that the Coulomb branch dimension of the mirror shown in figure. 13 is the same as the number in 3.9. The mirror theory does not have a pure Coulomb branch, as the adjoint matter on $U(1)$ quiver node decouples; One can give the vevs to these matter at the generic point of Coulomb branch, and the dimension of the Higgs factor is just $\frac{(n-1)(k-1)}{2}$, which is the same as the Coulomb factor of the mixed branch of original 4d theory.

3. In fact, the above proposal can be checked using the known information of Higgs branch of original 4d AD theory when $k = 1$. The idea is following: for the 4d theory, the Higgs branch is pure and is given by the following variety

$$\overline{\mathcal{O}_q} \cap S_f \tag{3.10}$$

One can find a 3d quiver whose Coulomb branch is the same as above variety using the brane construction [3, 36] (See [35] for the detailed computation of this example), and the result agrees with the mirror shown in figure. 13.

**Case** $k < 0$: The above proposal for the mirror quiver is only good for $k \geq 0$. To find the mirror for theories with $-n < k < 0$, we look at the AD matter again. The Higgs factor of the mixed branch of original 4d theory is given by $\overline{\mathcal{O}_q} \cap S_f$ [35, 37]. Here

$$\mathcal{O}_q = [\underbrace{n+k,\ldots,n+k}_{n_1},\underbrace{n+k,\ldots,n+k}_{a},b], \ \ f = [(n+k-1)^{n_1}, 1^{n+n_1}]; \tag{3.11}$$

and we have $a(n+k) + b = n$ with $b < (n+k)$. Notice that $a, b$ is uniquely determined by the pair of numbers $(n, k)$.

To find the 3d mirror, we use following strategy: firstly, we use brane construction of [3, 36] to construct a quiver so that its Coulomb branch is given by $\mathcal{O}_q \cap S_f$, see formula 3.11, and the quiver is given in figure. 14 (without the adjoint matter). Then we look at the Higgs branch of the mirror quiver, whose dimension should match the Coulomb branch dimension, see formula. 3.7. Similarly to our previous case, we conjecture that this requires the addition of extra adjoint matter on $U(1)$ node, and the number can be easily computed. The mirror then takes the form shown in figure. 14 [7].

---

[7]For $n+k = 1$, we have $z = 0$, but there are still $a$ quiver nodes with rank $n_1$ on the blue tail.

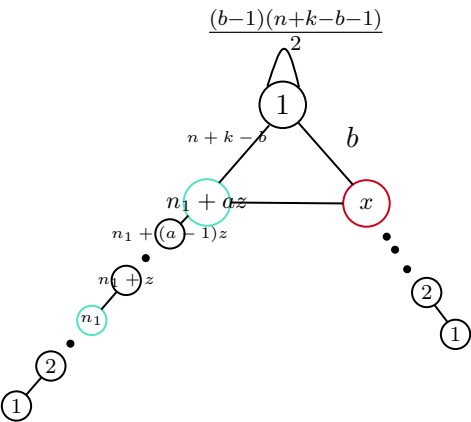

**Figure 14**. The 3d mirror for AD matter of type $(k, n)$ with $k < 0$. The flavor symmetry of AD matter is $U(n_1) \times SU(n + n_1)$. We have $a(n + k) + b = n$, with $b < (n + k)$. Here $x = n + n_1 - (a + 1)$, $z = n + k - 1$.

**Example**: Let's take $n + k = 2$, and the corresponding 4d theory has enhanced flavor symmetry $SU(n + 2n_1)$, and is actually equivalent to $D_2(SU(n + 2n_1))$ theory (This can be verified by comparing the Coulomb branch spectrum). The mirror of this theory was found in [38]. Now look at our quiver in figure. 14, we have $b = 1, a = \frac{n-1}{2}$ [8], and $z = 1, x = \frac{n-1}{2} + n_1$. Using these numbers, the quiver in figure. 14 is the same as that given in [38].

For general case, it is now easy to get the rules for finding the 3d mirror (assuming there is $m$ simplest black punctures, a maximal blue and red puncture): a) The quiver tail for red and blue puncture are the same as the bottom part of quiver. 14 (removing the $U(1)$ quiver node with adjoint matter), with the red and blue quiver nodes as the maximal ones, the parameters are changed as follows: $x = n_1 + \mathbf{m}n - (a + 1)$, $z = \mathbf{m}(n + k) - 1$; b): For each black puncture, there is a quiver tail, and the number of adjoints are modified as $\frac{(b-1)(n+k-b-1)}{2}$. Finally, there are $b$ edges between black and red nodes, and $n + k - b$ edges between blue and black nodes, and $b(n + k - b)$ edges between black nodes.

**Degenerating case**: Up to this point, we have only considered the simplest black puncture. We now verify our proposal for the degenerating case. The example is represented by a $(3, 2)$ type sphere with one trivial red puncture, and one trivial blue, and a black puncture of type $[1, 1]$. This theory is studied in [22] (see section 5 of that paper and it is claimed that this is the rank two $H_0$ theory), and its Coulomb branch has dimension 2. The 3d mirror for the theory is shown in figure. 15 and its Higgs branch indeed has dimension 2.

---

[8] $n, k$ coprime, and $n + k = 2$ implies that $n, k$ are both odd integers.

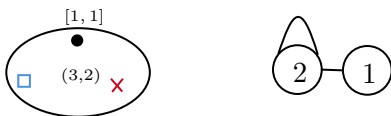

**Figure 15**. Left: a type $(3, 2)$ theory with one trivial red, one trivial blue and one black puncture with Young Tableaux $[1, 1]$; Right: the 3d mirror for the theory on the left.

## 3.2 $D_N$ type theory

Let's now discuss the 3d mirror for AD theories engineered using 6d D type $(2, 0)$ SCFTs. These theories are called $D_N$ type theories. For these theories, the flavor symmetry could be of $A, B, C, D$ type. Here we list the quiver tail whose Higgs branch has a flavor symmetry $G = ABCD$, see figure. 16. The corresponding 3d SCFT is called $T(G)$ theory [36]. The flavor symmetry group on the Coulomb branch is $G^L$, which is the Langlands dual group of $G$. These quiver tails would be useful for our studies later.

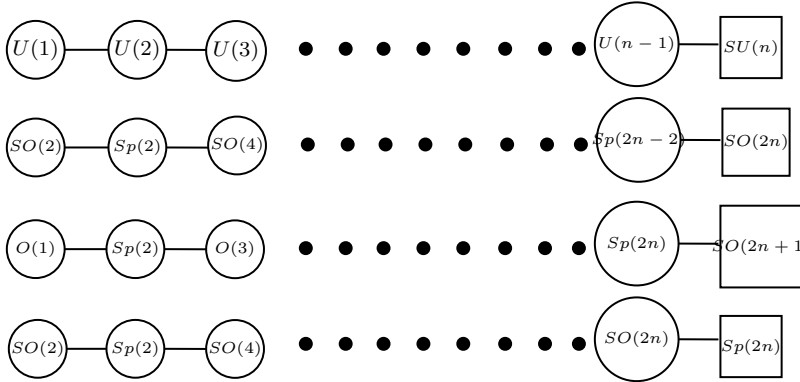

**Figure 16**. The quiver tails for 3d $T(G)$ theory, here $G$ is given by the group staying at the last square node. This class of theories has a $G$ type flavor symmetry on Higgs branch, and a $G^L$ (the Langlands dual group of $G$) flavor symmetry on Coulomb branch.

The $T(SU(N))$ theory and $T(SO(2N))$ type quiver tails are self mirror: the mirror quiver is the same as the original quiver. While $T(SO(2N + 1))$ type and $T(Sp(2N))$ type tails are mirror to each other. There is one more interesting self-dual mirror tail which would be also useful to us later, see figure. 17, and we call it $T(Sp^{'}(2N))$ theory.

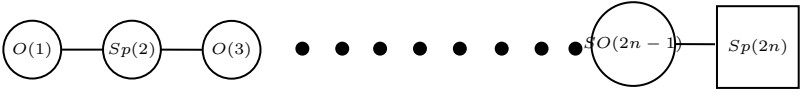

**Figure 17**. A quiver tail whose Higgs branch has flavor symmetry $Sp(2n)$, and its Coulomb branch also has flavor symmetry $Sp(2n)$.

Let's now discuss D type AD theories, which is constructed using 6d $D$ type $(2, 0)$ SCFT

on a sphere with one irregular and one regular puncture. We also need to consider theories constructed using outer automorphism [15]. They can be represented by a punctured sphere with three kinds of punctures: a red, a blue, and a black puncture; and there is a Young Tableaux for each puncture [9]. We do have an extra label $(p, q)$ for the punctured sphere. There are some differences with $A$ type theories though. Firstly, the blue and red punctures can be either $D$ or $C$ type. If the red puncture is of the $D$ type, it is called untwisted theory; and if the red puncture is of the $C$ type, it is called twisted theory. Secondly, they are two class of theories:

**Class I**: The first class of theories is labeled by a pair of co-prime integers $(k, n)$, here $n$ is even. There are some new features of this class of theories:

1. The red and blue puncture is of the same (opposite) type if there are even (odd) number of black punctures.

2. The black puncture with Young Tableaux [1] does not carry flavor symmetry.

Again, the size of Young Tableaux of these punctures are related as follows:

$$Y_{red} = Y_{blue} + n \sum_{black} Y_i \tag{3.12}$$

Here for a $C$ type puncture, $Y$ shifts by two: $Y = Y' + 2$ ($Y'$ the size of its Young Tableaux).

**Class II**: This class is labeled by a pair of integers $(2k, 2n)$, here $(k, n)$ is co-prime and $n$ is odd. The properties are:

1. The red and blue puncture is of the opposite (same) type if there are odd (even) number of black punctures, notice that the rule is opposite to that of the $(k, n)$ class.

2. Each black puncture of the type [1] carries a $U(1)$ flavor symmetry.

Again, the size of Young Tableaux of these punctures are related as follows:

$$Y_{red} = Y_{blue} + 2n \sum_{black} Y_i \tag{3.13}$$

Here for a $C$ type puncture, $Y$ shifts by two: $Y = Y' + 2$ with $Y'$ the size of its Young Tableaux.

To find 3d mirror of these theories, we follow the same strategy that we use for A type theories: We first assign a quiver tail for each puncture, and then find the gluing rules for them. The basic ideas are the same as what we did for A type theories, but because of the new features of $D$ type theories, the details are a lot more complicated.

Let's first consider 3d mirror for $D$ type class I theories, which are labeled by a pair of integer $(k, n)$ with $n$ even. The rules for assigning quiver tails to punctures are:

---

[9]A $C_N$ type puncture is labeled by a Young Tableaux $[n_1, \ldots, n_r]$ with $\sum n_i = 2N$, and **even** partition appear **odd** times; A $D_N$ type puncture is labeled by a Young Tableaux $[n_1, \ldots, n_r]$ with $\sum n_i = 2N$, and **even** partition appears **even** times.

1. For each $C$ type puncture, we assign a **B** type quiver tail (the end node has $B$ type flavor symmetry), see figure. 16. The reason is: $C$ type maximal puncture carries a $Sp$ type flavor symmetry acting on Higgs branch, which should be mapped to the Coulomb branch symmetry of the mirror quiver, which gives the $B$ type quiver.

2. For each $D$ type puncture, we assign a **D** type quiver tail (the end nod has $D$ type flavor symmetry).

3. For each $A$ type puncture, we assign a **A** type quiver tail. In particular, for the simple black puncture with label [1], the quiver is just a $U(1)$ node with certain number of adjoints which depend on $k$ and $n$. The method of determining the number is the same as the type $A$ case: by matching the dimension of the moduli space.

The quiver tails for various punctures are shown in figure. 18.

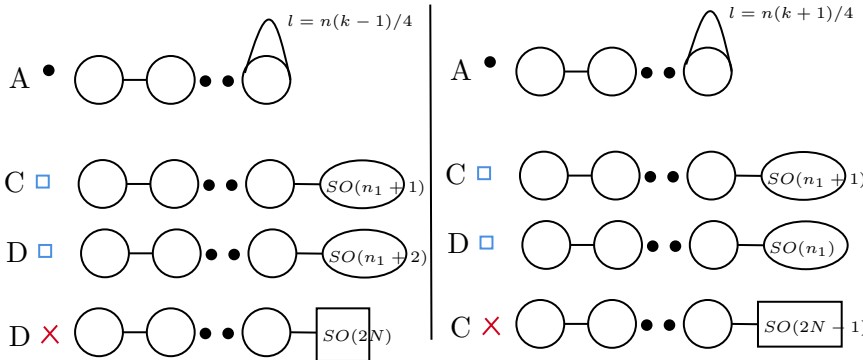

**Figure 18**. Left: Quiver tail for untwisted D type theories of type $(k, n)$; Right: Quiver tail for twisted D type theories of type $(k, n)$.

For the blue tail, we need to add some fundamental hypers to the last gauged node so that the Higgs branch of the quiver tail matches with the contribution of the blue puncture to the Coulomb branch of 4d theory. The rule is following: a): for the $D$ type blue puncture whose quiver tail has an ending node $SO(m)$, only a $SO(m-1)$ subgroup is gauged; b): if there is an odd number of black punctures, the number of hypermultiplets on the ending node is $\frac{k\sum n_i - 1}{2}$, here $n_i$ is the size of the black puncture; if there is an even number of black punctures, the number of hypermultiplets are $\frac{k\sum n_i - 2}{2}$.

The next step is the **gluing** rule, which is a lot more complicated comparing with that of type $A$ theories. It is possible to guess the rules by following computations: a) The flavor symmetry of the mirror quiver should math that of the original theory; b): The dimension of Higgs (Coulomb) branch of the mirror should be the same as that of Coulomb (Higgs) branch of the original theory [10]. After some computations, we find:

---

[10]There are some subtle points: sometimes the naive computation from the mirror does not match with that of the original theory.

1. For the $D$ type quiver whose end nodes are of the type $SO(2n)$, only a $SO(2n-1)$ subgroup is gauged.

2. There are $\frac{(k-1)}{2}$ edges between **blue** tail and **black** tail.

3. There are $\frac{n}{2}$ edges between **red** tail and **black** tail.

4. There is one edge between the **red** tail and **blue** tail. Here an edge represents a half-hypermultiplet.

Using above rules, we find the 3d mirror for a AD matter studied in [35] (labeled as class $B$ in that paper). This theory has flavor symmetry $Sp(n_1) \times SO(2N)$, and is represented by a $(k,n)$ type sphere: there is one red D type puncture, one blue C type puncture, and one black puncture. The 3d mirror is shown in figure. 19. The basic numeric relation between these integers is $2N = n + n_1 + 2$, and so our theory is specified by three integers $(k, n, n_1)$. One can make following checks for our mirror proposal:

1. The Coulomb branch dimension of original 4d theory is:

$$n_C = \frac{(n+k-1)(2n_1+n+1)}{4} \tag{3.14}$$

One can compute the Higgs branch dimension of the mirror quiver in figure. 19, which agrees with above number.

2. The Higgs factor of the mixed branch of original theory is $\overline{\mathcal{O}}_q \cap S_f$ [35], here $\mathcal{O}_q$ is a Nilpotent orbit of $D$ type, and $S_f$ is the Slodowy slice associated with the a nilpotent orbit $f$ . The data for two nilpotent orbits are

$$\mathcal{O}_q = [\underbrace{n+k, \ldots, n+k}_{n_1}, n+1, 1], \quad f = [(n+k-1)^{n_1}, 1^{n+n_1+2}] \tag{3.15}$$

One can compute the dimension of this variety, which is equal to the Coulomb branch factor of the mixed Coulomb branch of the mirror theory.

3. One can find the mirror quiver for $k = 1$ using brane construction introduced in [36]. In this case, $\mathcal{O}_q = [\underbrace{n+1, \ldots, n+1}_{n_1+1}, 1], \quad f = [n^{n_1}, 1^{n+n_1+2}]$. One find that the mirror is the same as that of figure. 19.

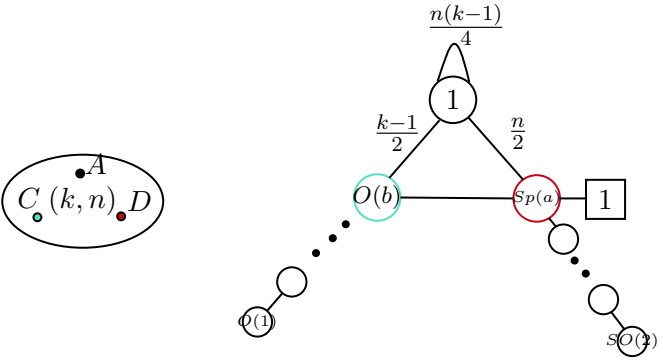

**Figure 19**. The flavor symmetry for the left theory is $SO(2N) \times Sp(n_1)$, this is the type $B$ AD matter studied in [35]. The 3d mirror is shown on the left. Here $a = 2N - 2, b = n_1 + 1, n + n_1 + 2 = 2N$. The left tail is a $TSO(n_1 + 1)$ (B type) tail, while the right one is a $TSO(2N)$ (D type) tail. Here the extra fundamental matter on $Sp(a)$ gauge group is a half-hyper, so there is no flavor symmetry on Higgs branch of this quiver. If $n_1 = 0$, so the blue puncture is trivial. The mirror is modified as follows: there are $\frac{(n+2)(k-1)}{4}$ adjoints on the $U(1)$ node.

**Example**: Let's use the brane construction to confirm our mirror proposal. We take $k = 1, n = 4, n_1 = 2$, and so the Higgs branch of the theory is given by $\overline{\mathcal{O}}_q \cap S_f$ with $\mathcal{O}_q = [5, 5, 5, 1], f = [4, 4, 1^8]$. To find a quiver whose Coulomb branch is given as $\overline{\mathcal{O}}_q \cap S_f$, we use the following method: we first find the dual partition of $\mathcal{O}_q$: $\mathcal{O}_q^D = [3, 3, 3, 3, 3, 1]$; and then we construct a $NS5 - D5 - D3$ configuration: on the left we have a $D5 - D3$ configuration which is determined by $\mathcal{O}_q^D$, and on the right we have a $NS5 - D3$ configuration which is determined by $f$, see figure. 20. Notice that here we need to use half-D5 and half NS5 brane. To find the quiver gauge theory from the brane description, we do the brane moves using the rule introduced in [36]. Finally, we find the configuration in figure. 20. The quiver read from the final configuration of figure. 20 is given there, and agrees with our proposal in figure. 19.

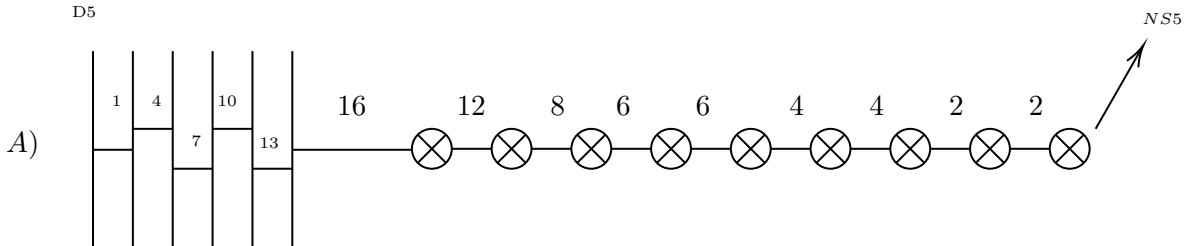

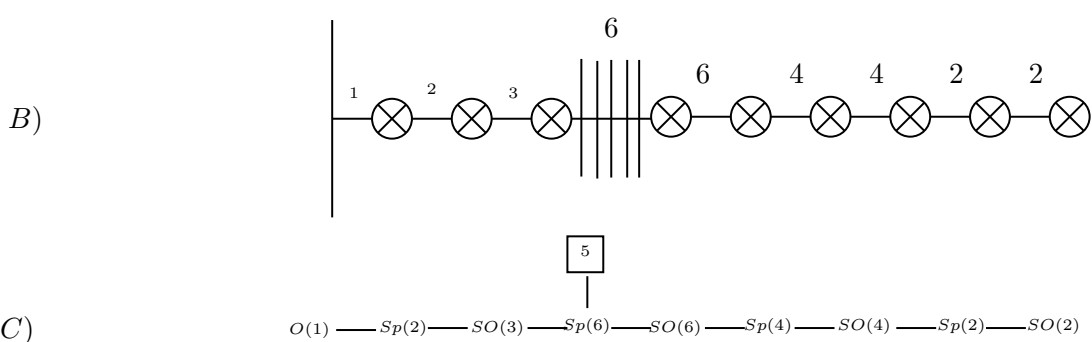

**Figure 20**. A): The brane configurations whose Coulomb branch would be $\overline{\mathcal{O}}_q \cap S_f$, here $\mathcal{O}_q = [5,5,5,1]$ and $f = [4,4,1^8]$, and they are both nilpotent orbits of $D_8$ algebra. B): To find a quiver description, we move the D5 brane according to the rule given in [36]; C): The quiver description from the brane configuration in part $B$.

Similarly, one can find the 3d mirror for AD matter with flavor symmetry $Sp(2N-2) \times SO(n_1)$ (labeled as class C theory in [35]). This class of theory is represented by one red $C$ type puncture, and one blue $D$ type puncture, and one simplest black puncture, see figure. 21. The 3d mirror of this class of theories is shown in figure. 21. One can make similar checks as we did before, and we leave the details to interested reader.

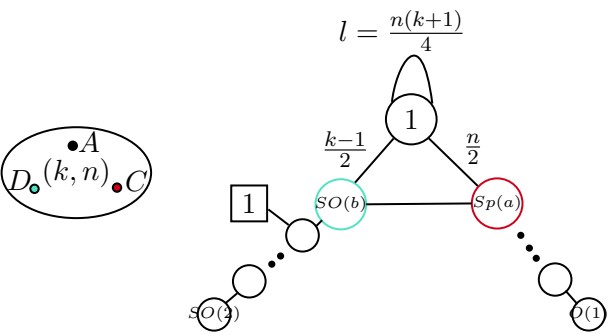

**Figure 21**. The flavor symmetry for the left theory is $Sp(2N-2) \times SO(n_1)$, which is the type C AD matter studied in [35]. We have $n + n_1 = 2N$, both $n$ and $n_1 > 2$ are even. The 3d mirror is shown on the left. The numbers $a$ and $b$ are given as $a = 2N - 2, b = n_1 - 1$. The left tail is a $TSO(n_1)$ tail (D type tail), while the right one is a $TSO(2N-1)$ tail (B type tail). If $n_1 = 0$, now $b = -1$, but if we formally subtract the extra fundamentals to the adjoints of $U(1)$, we would get the right answer. So in this case, the number of adjoints on $U(1)$ is changed to $\frac{n(k+1)}{4} - \frac{1}{2}(k-1)$. If $n_1 = 2$, we change the $SO(1)$ node to be a $U(1)$ node, and the adjoints on $U(1)$ is changed to $\frac{(n-2)(k+1)}{2}$, moreover the number of edges between these two $U(1)$s is changed to $k + 1$.

If there are more than one simple black punctures, the 3d mirror is more subtle to find. First, we can not assign a $U(1)$ quiver node for each black puncture, as there is no flavor symmetry associated with black punctures. To find the mirror, we follow the following strategy: these theories has weakly coupled gauge theory description: it is decomposed into AD matter coupled with gauge groups. Since we know the 3d mirror for the AD matter, the 3d mirror can be found by gluing the 3d mirror of the AD matter. Here the new feature is that we need to use Coulomb branch gluing.

The implementation of this idea is shown in figure. 22. We start with a fourth punctured sphere with one maximal red D type puncture, one maximal blue D type puncture, and two simple black punctures. This theory has a gauge theory description [23], which is interpreted as degenerating fourth puncture sphere into two three punctured spheres. The gauge theory description can be interpreted from gluing 3d mirrors of the quivers for two three punctured spheres, see figure. 22. The glued quiver has two $U(1)$ quiver nodes. To match the flavor symmetry of original theory, we merge the $U(1)$ node, and finally, the mirror is shown in the bottom of figure. 22. The merging can be understood from matching Coulomb and Higgs branch dimension of the mirror to that of original 4d theory, see the discussion on the Coulomb branch gluing in next section.

**Remark**: The gauge group at the top of figure. 22 is of the C type, and the gluing works perfectly. If the gauge group is of the $D$ type, the gluing of the mirror quiver for the AD theory is more complicated: firstly one need to add more adjoints on the $U(1)$ node; secondly each D type gauge group in original 4d theory would create an issue in matching the Coulomb branch dimension of the mirror with the Higgs branch of the original theory. These two issues appear in the class $\mathcal{S}$ theory too. The second problem can be solved as follows: $Higgs_{4d} = Col_{mirror} + x$, here $x$ is the number of $D$ type gauge groups in the 4d

theory.

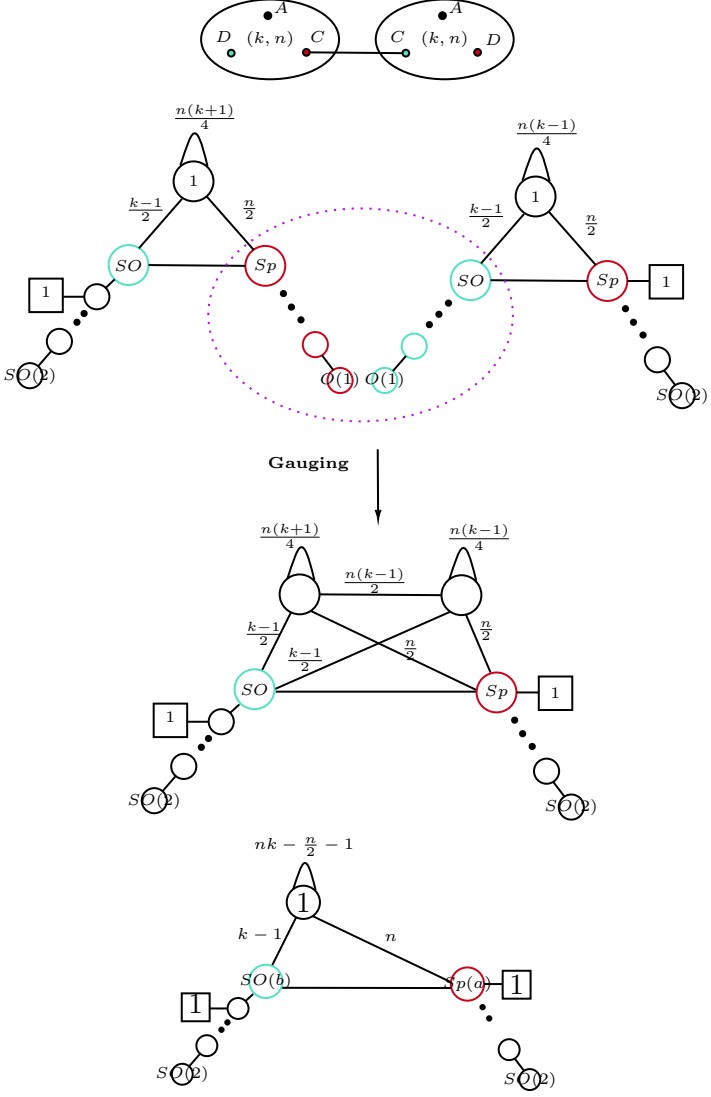

**Figure 22**. Up: A fourth punctured sphere is decomposed into two three punctured spheres, and this is the S-duality picture found in [23]; Middle: The gauge theory description is interpreted as gluing mirror quivers of two three punctured sphere [13]; Bottom: We merge the $U(1)$ nodes to get the 3d mirror for the fourth punctured theory; The number of adjoints are calculated as follows: first we sum the number of adjoints of previous two nodes, and add the number of bi-fundamental between them, finally, we subtract the previous number by one as the rank of gauge group is reduced by one after we merge two $U(1)$s.

For the general case, the 3d mirror should take the form shown in figure. 23. Here the polynomial $f(a, k, n)$ can be computed by matching the Higgs branch dimension of the mirror with that of the Coulomb branch dimension of the original 4d theory (this can be easily computed using the method developed in [28]). We leave the details to the interested

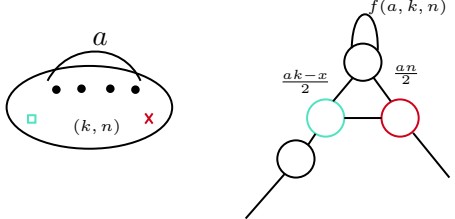

**Figure 23**. The 3d mirror for general D type theory of class $(k, n)$. We assume the blue and red puncture has maximal flavor symmetry, and the black puncture is the simplest one. The theory depends on the data $(a, k, n, n_1)$. If $a$ is even (odd), the red and blue puncture are of the same (different) type. Right: the 3d mirror for the theory listed on the right, and $f(a, k, n)$ is a polynomial which can be computed using the Coulomb branch data of original 4d theory. $x = 1$ if $a$ is odd, and $x = 2$ for even $a$.

reader.

Finally, let's briefly consider 3d mirror for class $II$ theory with label $(2k, 2n)$ (Here $n$ is odd, and $(n, k) = 1$). The assignment for a quiver tail for the blue and red punctures are the same as class $I$ theory. The major difference here is that there is a $U(1)$ flavor symmetry for a single black puncture. The naive guess is to assign a quiver with two $U(1)$ nodes for a simple black puncture. To figure out the edges between these two nodes and the number of adjoints on the $U(1)$ node, We choose special value for $n$ and $k$ so that the theory can be engineered using other 6d $(2,0)$ theory, from which we can read the 3d mirror using previous results.

Consider three punctured sphere with trivial (untwisted) red and blue puncture, and this theory can be engineered by the type IIB string theory on the singularity $x^2 + y^{n-1} + yz^2 + zw^k = 0$ (The Coulomb branch dimension of this class of theory is $\frac{1}{2}(2kn - 2k - n - 1)$). We have following isomorphisms:

- If we take $n = 3$, the singularity takes the form $x^2 + y^2 + z^4 + zw^k = 0$, then this theory is equivalent to $A_3$ theory on a punctured sphere with label $(k, 3)$, and blue puncture with label $[1]$, the red and black puncture is trivial. and we know its mirror: which has one two $U(1)$ quiver nodes with $k$ edges connecting them, and there is also a $k - 1$ adjoints on the other $U(1)$ node.

- If we take $k = 1$, then this theory is equivalent to $(A_1, A_{n-2})$ theory ($n$ is odd), and its mirror has two $U(1)$ nodes with $\frac{n-1}{2}$ edges connecting them.

To fit above data, we have the following mirror for the simple black puncture of $(2k, 2n)$ theory (with trivial blue and red puncture). One can check that this gives the correct Higgs branch and Coulomb branch dimension based on the prediction of 3d mirror symmetry.

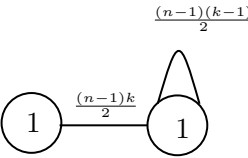

**Figure 24**. 3d mirror for the theory engineered by the singularity $x^2 + y^{n-1} + yz^2 + zw^k = 0$.

We now consider the theory which is represented by a sphere with one simplest black puncture, one trivial (untwisted) red puncture, and one simple blue puncture (with $SO(2)$ flavor symmetry) (type $(2k, 2n)$ theory). This theory is engineered by the singularity $x^2 + y^n + yz^2 + w^{2k} = 0$ (The Coulomb branch dimension is $\frac{1}{2}(2kn + 2k - n - 3)$). We have following isomorphism:

- If we take $k = 1$, then this theory is equivalent to $(A_1, D_{n+1})$ theory. Its mirror is given by two $U(1)$ nodes with $\frac{n-1}{2}$ edges connecting them, and an extra $U(1)$ node connected with above two $U(1)$ quiver node with a single edge.

- If we take $n = 1$, then this theory is equivalent to two copies of $(A_1, A_{2k-1})$ theory, and one knows the mirror of it: the mirror is given by two $U(1)$ node with $k$ edges between them.

To fit above data, we have the following mirror for the simple black puncture of $(2n, 2k)$ theory (with a simple blue puncture and a trivial red puncture).

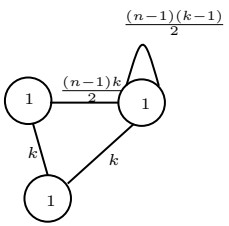

**Figure 25**. The 3d mirror for theory engineered by the singularity $x^2 + y^n + yz^2 + w^{2k} = 0$.

Given above two class examples, we can now figure out the quiver for the simple black puncture, which is given by the quiver in figure. 24. Once we know the quiver tails for all three types of punctures, we can find out the 3d mirror for general class II theories. We show an example in figure. 26. If there are more than one black punctures, one can find its 3d mirror by using the gluing method of the 3d mirror corresponding to three punctured sphere.

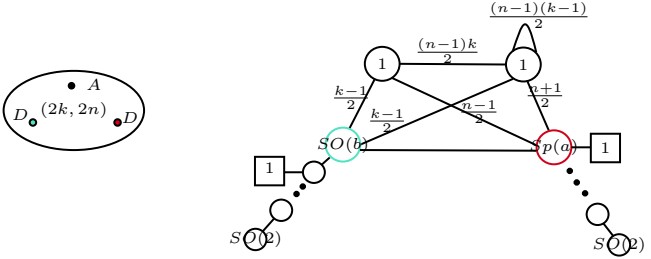

**Figure 26**. The flavor symmetry for the left theory is $SO(2N) \times SO(n_1) \times U(1)$. Here $n_1$ is even and $2n + n_1 = 2N$, we also take $k$ odd. The 3d mirror is shown on the left. Here $a = 2N-2, b = n_1 - 1$. The left tail is a $TSO(n_1)$ tail (only a $SO(n_1 - 1)$ subgroup is gauged), while the right one is a $TSO(2N)$ tail. The addition of a blue puncture (the flavor symmetry is $SO(n_1)$ change the Coulomb branch dimension of 4d theory by the number $-k + kn_1 - \frac{n_1^2}{4}$, which is accounted by the blue quiver tail, and there should be a total of $k-1$ bi-fundamental hypermultiplets on the blue $SO(b)$ quiver node. Here we guess the rule for the gluing of the blue tail to that of the black puncture, and it would be interesting to find other ways to verify it.

## 3.3 Twisted $A_{2N-1}$ and $A_{2N}$ type theory

Let's now consider 3d mirror for twisted $A$ type theories [15]. Here we consider the class of theories labeled by pair $(k + \frac{1}{2}, n)$, and $n$ is constrained to be odd. For these theories, we have $B$ type and $C$ type punctures (blue or red). The black puncture is of the $A$ type. The quiver tail for the punctures are:

1. The quiver tail for $B$ type puncture is $T(Sp(2n))$ tail.

2. The quiver tail for $C$ type puncture is different: we attach the self-dual quiver tail $T(Sp'(2n))$ theory for it.

3. For the simple black puncture, we attach a $U(1)$ quiver tail with certain number of adjoints (The number can be figured out by matching with the Coulomb branch data of the 4d theory [15]).

The gluing rule is also similar to theories considered above, here we do not try to repeat them. We only list some mirror pairs in figure. 27 and 28. These theories are important as they are the basic AD matter, which can be used to build more complicated conformal theories.

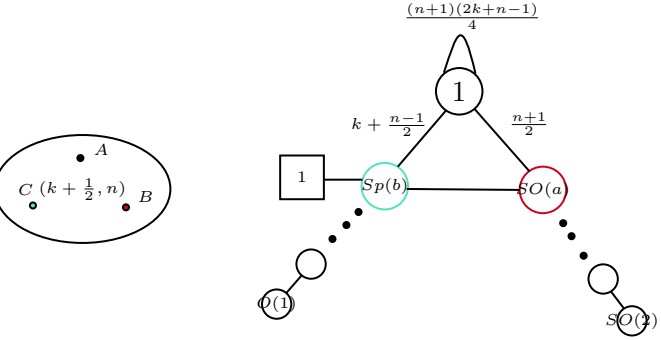

**Figure 27**. Here $n$ is odd and $n_1$ is even, and the flavor symmetry is $SO(2N+1) \times Sp(n_1)$. We have $n + n_1 + 1 = 2N$. Here $b = n_1$ and $a = 2N$.

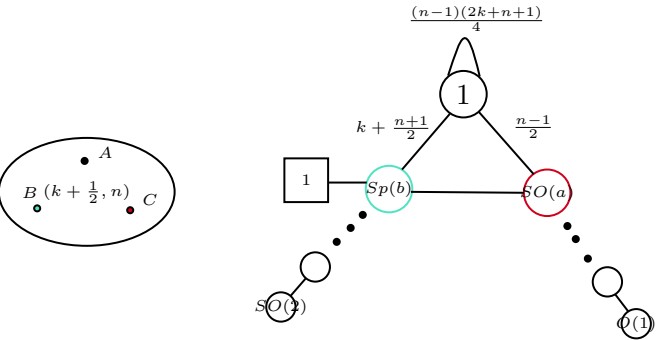

**Figure 28**. Here $n$ is odd and we have $n + n_1 = 2N$. The flavor symmetry for the left theory is $Sp(2N) \times SO(n_1 + 1)$. The 3d mirror is shown on the left. Here $a = 2N - 1, b = n_1$. The left tail is a $T Sp(n_1)$ tail, while the right one is a $T^{'} Sp(2N)$ tail.

To find the mirror for more general theories, we can use the similar strategy that we have done for D type theories: i.e. the 3d mirror for more general case is found by gluing the quiver of the AD matter.

### 3.4 Exceptional type theories

Let's now discuss the 3d mirror for 4d $\mathcal{N} = 2$ SCFT engineered using 6d E type $(2, 0)$ theory. While in general we could not find a quiver gauge theory description, we can use known components and get the mirror by gluing construction. The basic building block is the $T(G))$ theories constructed in [36]. Although these theories do not have a Lagrangian description, we still know many of their properties:

1. The Higgs branch is identified with closure of the maximal nilpotent orbit of the Lie algebra $\mathfrak{g}$ associated with $G$. The Coulomb branch is identified with the closure of maximal nilpotent orbit of $G^{\vee}$, here $G^{\vee}$ is the Langlands dual of $G$. The compelx dimension of the moduli space is given by $dim(G) - rank(G)$.

We can again represent our theory by a sphere with three kinds of punctures, and an extra label $(k, n)$. For the blue and red puncture, we assume it takes the maximal form. The theory for the blue and red marked puncture is given by $T(G^\vee)$ if the flavor symmetry is $G$. The quiver for the black puncture can be figured out using the Coulomb branch data of original theory. For the simplest black puncture, it is simply a $U(1)$ gauged node with certain number of adjoints. The gluing rule is again similar to what is discussed above. Here we give an example in figure. 29. The interested reader can work out the mirror for other cases.

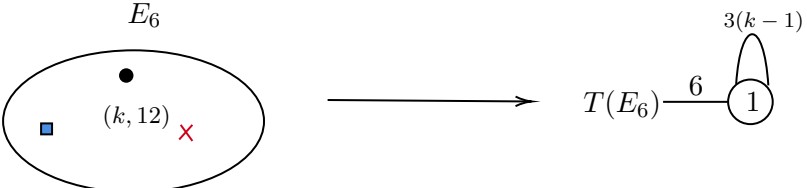

**Figure 29**. Left: a $E_6$ type theory which is represented by a trivial blue and a maximal red puncture (with $E_6$ flavor symmetry), and the black puncture is simple, here $k$ is co-prime with 12. The physical data for the theory on the left (we take $k > 0$): The Coulomb branch has dimension $n_c = 3k + 33$; The mixed branch has dimension $n_h = 36, n_c = 3k - 3$. The mirror theory is shown on the left: here we gauge $U(1)^6$ subgroup of $E_6$ flavor symmetry diagonally, and there are $3k - 3$ adjoints on $U(1)$ node. The mixed Coulomb branch of the mirror theory has dimension: $n_c = 36, n_h = 3k - 3$, which matches with the mixed Higgs branch of the original theory; and the Higgs branch of the mirror theory is $3k - 3 + 36 = 3k + 33$, which is the same as the Coulomb branch of original 4d theory.

## 4 Some applications

In this section, we will give several applications of the 3d mirror found above.

### 4.1 Higgs branch of 4d AD theory

The low energy physics on Coulomb branch of 4d AD theory is nicely described by the spectral curve of the Hitchin system, which can be written down explicitly if 6d configuration is given [10]. The Higgs branch is more complicated and typically it consists of a Higgs branch component and an interacting SCFT which does not have a Higgs branch [24]. One method of identifying the Higgs branch is through the 4d $\mathcal{N} = 2$/2d vertex operator algebra (VOA) correspondence. The Higgs branch component was found in [24] if one can find the associated 2d vertex operator algebra: the Higgs branch is identified with the associated variety of the VOA [12, 39]. This method has some limitations: a): We only know the VOA for a subset of AD theories; b): Even if we have the description of VOA, it is difficult to find the associated variety. Here by using 3d mirrors found above, we can evade above difficulties and found the Higgs branch of original 4d theories.

According to the basic map of 3d mirror, the Coulomb branch of 3d mirror SCFT is equal to the Higgs branch of original 4d AD theory. Since the Higgs branch of 4d theory has a Higgs component and a Coulomb component, the Coulomb branch of 3d mirror should have a Coulomb and Higgs component. Once we know 3d mirror, we can study its mixed Coulomb branch and then find the mixed Higgs branch of original 4d theory.

Let's illustrate this point using the type $A$ theory. We start with a 6d $A_{N-1}$ $(2,0)$ SCFT on a sphere with following irregular singularity:

$$\Phi = \frac{T}{z^{2+\frac{ak}{an}}} + \dots \tag{4.1}$$

Here $(n, k) = 1$, and $T$ is regular semi-simple element, and $an = N$. This is also called $(A_{ak-1}, A_{an-1})$ theory. This theory can be represented by a sphere with $a$ simple black puncture, one trivial blue and one trivial red puncture, see figure. 30. The 3d mirror is shown in figure. 30. Using the proposal of 3d mirror in last section, we find that: the 3d mirror has **a** $U(1)$ quiver nodes, and each $U(1)$ quiver node has $l = \frac{(n-1)(k-1)}{2}$ adjoints (which is in trivial representation for $U(1)$ gauge group), there are also $nk$ edges between the $U(1)$ quiver node, see figure. 30. The 3d mirror suggests the Higgs branch of original AD theory has two components:

1. A Higgs component which is described by the Coulomb branch of the quiver with $a$ $U(1)$ quiver nodes and $nk$ edges between them. An overall $U(1)$ of the mirror quiver is decoupled, so the Coulomb branch dimension of the mirror theory is just $a - 1$, which gives the dimension of the Higgs component of the Higgs branch of original 4d AD theory.

2. The Coulomb branch of 3d mirror has $\frac{a(n-1)(k-1)}{2}$ free hypers. This suggests that the Higgs branch of original 4d theory has an interacting theory part which do not have a Higgs branch. The structure of 3d mirror implies that the the interacting theory consists of $a$ copies of $(A_{n-1}, A_{k-1})$ theory.

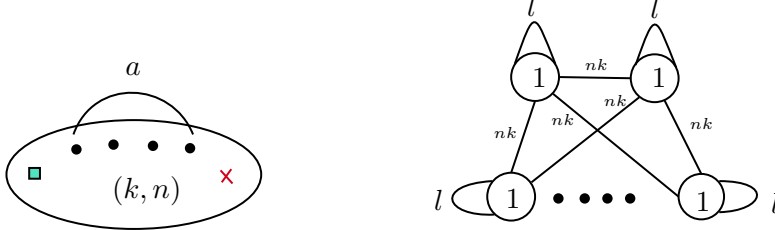

**Figure 30**. Left: punctured sphere representation of $(A_{ak-1}, A_{an-1})$ theory, there are $a$ simple black punctures, one trivial red puncture and one trivial blue puncture. Right: 3d mirror for $(A_{ak-1}, A_{an-1})$ theory: there are $a$ $U(1)$ quiver nodes, and $nk$ edges between each pair of $U(1)$ node. Here $l = \frac{(n-1)(k-1)}{2}$.

Let's verify above proposal of Higgs branch of 4d theory by using anomaly matching of $(a-c)$. First, we can compute $a-c$ using Coulomb branch data of 4d AD theory [28], and we express it in terms of a growth function as $-\frac{G}{48} = a - c$. The anomaly matching condition [24] is

$$G_{UV} = 2n_h + G_{IR} \tag{4.2}$$

Here $G_{IR}$ is the growth function of IR interacting theory. Let's verify our proposal by looking at the theory with $a = 2, n = 2, k = 3$ (($A_{5,3}$) theory), and the mixed Higgs branch of 4d theory should have $n_h = 1$ and the IR theory has 2 copies of $(A_1, A_2)$ theories, which is derived using 3d mirror. We find $G_{UV} = \frac{14}{5}$, $n_h = 1$, and $G_{IR} = 2 \times G_{(A_1, A_2)} = \frac{4}{5}$, and the above equality is correct! One can verify that the above equality is always true for the above stated Higgs branch proposal.

## 4.2 Coulomb branch gluing

One can get new 3d $\mathcal{N} = 4$ theory by performing a Higgs branch gluing: we start with two matter systems, and both of them has flavor symmetry group $G$ acting on Higgs branch; and we can form a new theory by gauging diagonally the symmetry group $G$. Mathematically it is just the hyperkahler quotient studied in [40], which gives the Higgs branch of the glued theory. If our theory admits a quiver gauge theory description, the Higgs branch gluing is quite simple, see figure. 31. The Coulomb branch and Higgs branch dimension of the glued theory is following

$$d_C = Col_1 + Col_2 + N - 1, \quad d_H = Hig_1 + Hig_2 - (N^2 - 1) \tag{4.3}$$

here $Col_i, i = 1, 2$ are the Coulomb branch dimension of the matter, and $Hig_i, i = 1, 2$ are the Higgs branch dimension of the matter. Here we assume that we gauge a $SU(N)$ flavor symmetry of two quivers (diagonal gauging) [11]. The Higgs branch of the gauged system is given by the hyperkahler quotient of original two systems, and the Coulomb branch of gauged system is more complicated and does not have simple description.

---

[11]More generally, if we gauge diagonally a simple group $G$ of Higgs branches of two matter systems, we have $d_C = Col_1 + Col_2 + rank(G), \quad d_H = Hig_1 + Hig_2 - dim(G)$.

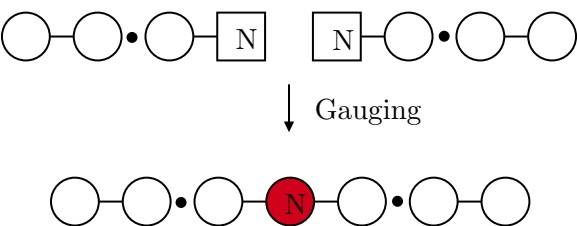

**Figure 31**. Higgs branch gauging: here we diagonally gauge $SU(N)$ flavor symmetry of two quivers

In this paper, the gluing we use is the Coulomb branch gauging, namely, the flavor symmetry which is gauged acts on Coulomb branch. The corresponding current of the flavor symmetry on the Coulomb branch can be constructed using monopole operators, see [36]. Here we use a simple result for the quiver gauge theory with unitary gauge groups: for a balanced [12] subquiver of ADE type, the flavor symmetry would be the corresponding ADE type.

The Coulomb branch gluing is the mirror of Higgs branch gluing, see figure. 32. Let's consider the coulomb branch gluing for the quivers, namely, we would like to glue two quivers to form a new quiver such that the Coulomb branch of the new quiver is the hyperkahler quotient of the Coulomb branch of original quivers! In particular, we would like to ensure that the Coulomb branch dimension of the glued quiver is given by the following formula

$$d_C^{mirror} = Col_1 + Col_2 - (N^2 - 1) \tag{4.4}$$

(here we assume that we gauge a $SU(N)$ flavor group). To match the Higgs branch gluing, we also require that the Higgs branch of the combined system has the following dimension formula

$$d_H^{mirror} = Hig_1 + Hig_2 + (N - 1) \tag{4.5}$$

It is this requirement which makes the Coulomb branch gluing much more complicated. Amazingly, such a procedure is possible, and see figure. 32 for an example.

Let's verify that the Coulomb branch and Higgs branch dimension formula in 4.4 and 4.5. In figure. 32, we gauge a $SU(n_1)$ group on the Coulomb branch. The change of Coulomb branch dimension of the glued quiver is

$$\delta d_C = -[\frac{n_1^2 - n_1}{2} + (\frac{n_1^2 - n_1}{2} + n_1) - (1)] = -(n_1^2 - 1) \tag{4.6}$$

The origin of three terms in the equation is: the first term is due to the elimination of the circled subquiver of the left quiver, and the second term is due to the circled subquiver on the right (notice an extra $n_1$ contribution; the extra $-1$ is due to the fact one need to subtract an overall decoupled $U(1)$ for the quiver). This agrees with the expectation in equation 4.4. The change of Higgs branch dimension of the combined mirror quiver is the following

$$\delta d_H = -(\frac{n_1^2 - n_1}{2}) - (\frac{n_1^2 - n_1}{2} - n_1^2) - (a + k)n_1 + (bk + ab + an + nk) + 1 \tag{4.7}$$

---

[12]The number of flavors on a quiver node of this subquiver is equal to twice of rank: $N_f = 2N_c$.

Here the first term comes form left circled subquiver, and the second term comes form right circled subquiver, the third term is the hypermultiplets attached on the $n_1$ node of the circled quiver tail; the fourth term comes from added hypermulteplets in the glued quiver, finally there is an extra 1 comes from the overall decoupled $U(1)$s. Using the relation $n + b = n_1$, we find

$$\delta d_H = n_1 - 1 \tag{4.8}$$

which gives the desired formula, see 4.5.

The same computations can be done for ortho-symplectic quivers studied in last section, and we leave the details for the interested reader.

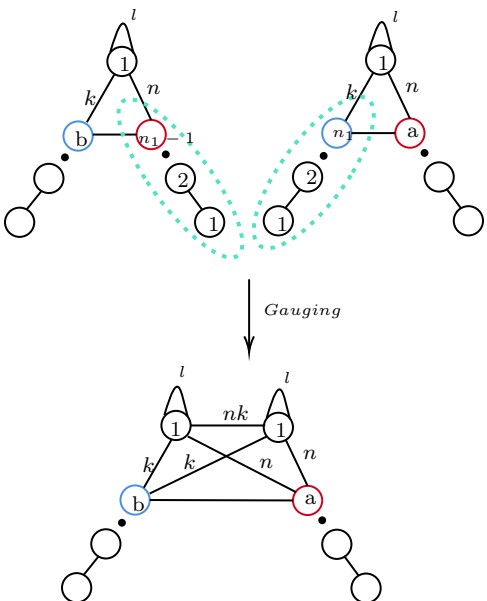

**Figure 32**. Up: Diagonal gauging $SU(n_1)$ flavor symmetry on Coulomb branch of two quivers, here $n + b = n_1$ so that the left circled sub-quiver carries $SU(n_1)$ flavor symmetry. Bottom: The gauging is achieved by eliminating two sub-quiver carrying flavor symmetry $SU(n_1)$ on the Coulomb branch, and 1): connecting $nk$ edges between two $U(1)$ nodes, 2) one edge between blue and red node, 3) $k$ edges between blue node and $U(1)$ node; 4) $n$ edge between red node and $U(1)$ node.

## 4.3  S duality for AD theory

Some 4d AD theories admit exact marginal deformations and it is an interesting question to find its weakly coupled gauge theory descriptions. For a given AD theory, it is possible to find more than one such description, and these theories are S-dual to each other. The S-duality property of some AD theories were found in [13] by using 3d mirror: we start with the 3d mirror of AD theory and decompose it into various pieces representing AD matter, the weakly coupled gauge theory description is found from this decomposition. The gauging process is interpreted as gluing quiver tail (Coulomb branch gluing) in the 3d mirror picture. This method works for theories whose 3d mirrors were found [10]. More generally, the S duality property is found by using an extra punctured sphere representing

our AD theory, and weakly coupled theory is found by finding pants decomposition of it [22, 23].

Since we now has the 3d mirror for other AD theories, we would like to confirm the S-duality found in [22, 23] by using the decomposition of 3d mirror. Here we give a simple example, and the general case is quite similar. We consider $(A_3, A_5)$ theory which can be engineered by putting 6d $A_3$ theory on a sphere with following irregular singularity: $\Phi = \frac{T}{z^{2 + \frac{6}{4}}}$, and the regular singularity is trivial. It is a $(3, 2)$ type theory, and can be represented by a sphere with four punctures (two simple black punctures, one trivial red puncture and one trivial blue puncture). The weakly coupled gauge theory description is found by decomposing fourt punctured sphere into two three punctured spheres [13].

Now we would like to understand this operation in terms of 3d mirror. We start with the 3d mirror of the original theory (see figure. 33), and decompose it into two subquivers. These two subquivers representing the AD matter. The sub-quiver indeed gives the 3d mirror for the AD matter represented by two three punctured spheres. The use of 3d mirror to study S-duality has some nice consequences, i.e. the 3d mirror and its decomposition in figure. 33 shows that there is only one weakly coupled gauge theory description of this theory.

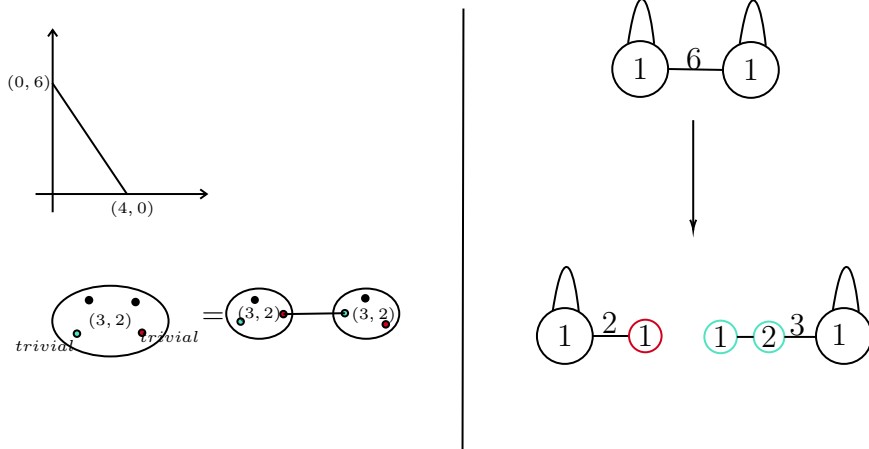

**Figure 33**. Left: punctured sphere representation for $(A_3, A_5)$ theory: it is a fourth punctured sphere with two simple black punctures, one trivial blue and one trivial red puncture. Right: We start with the 3d mirror of the original theory (which can be read from the fourth punctured sphere); the full quiver is decomposed into two sub-quivers glued together; each subquiver represents the mirror of AD matter described by two three punctured spheres.

## 4.4  New 3d $\mathcal{N} = 4$ SCFTs

We can get a large class of new $\mathcal{N} = 4$ SCFTs by compactifying 4d AD theory on a circle. Among these 3d $\mathcal{N} = 4$ SCFTs, there are a very interesting class of new theories, which could be thought of the generalization of bi-fundamental matter. These theories can be used to build a large class of new 3d $\mathcal{N} = 4$ SCFTs. These new matters are the dimensional

reduction of four dimensional AD matter, and the interesting feature is that they carry two non-abelian flavor symmetries. In the literature, we already know following SCFT which has interesting flavor symmetries on the Higgs branch:

1. $T(G)$ theory [36]: these theories have a Higgs branch and a Coulomb branch. The flavor symmetry on the Higgs branch is $G$ while the flavor symmetry on the Coulomb branch is $G^\vee$. This theory can be constructed using the boundary condition of 4d SYM theory.

2. $T_N$ theory and its $ADE$ generalization [21, 41]: $T_N$ theories have a Higgs branch with flavor symmetries $SU(N) \times SU(N) \times SU(N)$ (The $ADE$ generalizations have flavor symmetry group $G \times G \times G$). The Coulomb branch does not have flavor symmetry. This theory is constructed using the dimensional reduction of 4d $T_N$ theory

We can use these matter to construct quiver gauge theory and in the IR one find 3d $\mathcal{N} = 4$ SCFTs. In this paper, we found an infinite class of new 3d $N = 4$ SCFTs (labeled by pair of integers $(k, n)$), and they are represented by a sphere with one black puncture, one blue puncture, one one red puncture, see figure. 34. The flavor symmetry on Higgs branch is $SU(N_1) \times SU(N_2) \times U(N_3)$ (for A type theory). One could use these matters to construct new interesting 3d $\mathcal{N} = 4$ SCFT (some of them can be described as the dimensional reduction of 4d $\mathcal{N} = 2$ AD theories). More interestingly, one could use these matter to construct new $\mathcal{N} = 4$ Chern-Simons matter theory, and the results would appear in a separate publication.

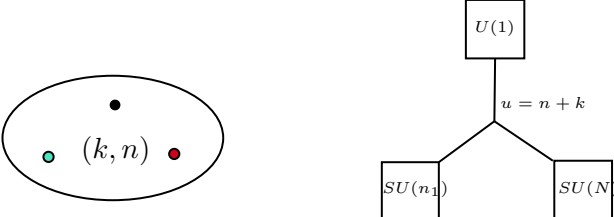

**Figure 34**. Left: 4d AD matter which has flavor symmetry $U(n_1) \times SU(N) \times U(1)$, and it is represented by a sphere with one red puncture, one blue puncture, and one simple black puncture. When $u = n + k = 1$, the theory is just bi-fundamental matter, so this class of theory can be thought of as the generalized bi-fundamental matter.

## 5 Conclusion

We found the 3d mirror for the 3d SCFT derived by compactifying 4d Argyres-Douglas theory on a circle. These AD theories are constructed using 6d $(2,0)$ SCFTs on a sphere with one irregular and one regular singularity. The crucial insight is that one can represent AD theory by a different punctured sphere, which is ued to find the S duality property of the AD theory, and in this paper we show that it is also very useful to find the 3d mirror of the AD theory.

The punctured sphere representation of the AD theory has three types of punctures, which we called blue, red, and black. The method of finding 3d mirror for the AD theory is following: first we attach a quiver tail for each puncture; then we glue these quiver tails together. It is relatively easy to find the quiver tail for each puncture, and it is rather difficult to find the gluing rule. We use various consistent checks to find the gluing rule, and therefore find the 3d mirror for all the AD theories constructed using 6d $(2,0)$ theory. The results are also consistent with all the known examples in the literature [17–20].

The 3d mirror is very useful to study the properties of original 4d AD theory: a) One can find the Higgs branch of the 4d theory by studying the Coulomb branch of the mirror theory. The Higgs branch of the 4d theory is in general difficult to obtain and so the 3d mirror is a very useful tool for this purpose; b) One can use the decomposition of 3d mirror to find the weakly coupled gauge theory description, and this has some advantage over the method used in [22, 23].

The quiver gauge theory found in this paper seems to have interesting applications in the study of purely 3d theories, and it would be interesting to further study them, i.e compute its Hilbert series on the Higgs and Coulomb branch. Moreover, we have found a large class of new 3d mirror pairs, and we believe the study of them would help us understand better 3d mirror symmetry.

It would be also interesting to study 3d mirror for other 4d $\mathcal{N} = 2$ SCFT constructed using three-fold singularity [28].

Our work is motivated in understanding the related mathematical work [42], but our construction is mainly based on physical considerations and it would be interesting to compare our results with [42].

## Acknowledgements

The author would like to thank P. Boalch and W.B Yan for helpful discussions. DX is supported by Yau mathematical science center at Tsinghua University.

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
