# Peer review of "d mirror for Argyres-Douglas theories"

_SciPost Physics_

## Round 1 · Referee Report · Anonymous (Referee 1) · 2024-5-22

Strengths

1- The draft provides a uniform treatment of the 3D mirrors of all AD theories obtained by compactifying the 6D (0,2) ADE theories.

Weaknesses

1- The scope of validation is unclear. Major conjectures and hypotheses have only been demonstrated in previously worked-out cases. In most other instances, only dimensions and global symmetry algebras are matched.

2- The presentation could be improved. It is challenging to extract the results from the paper as currently written.

3- The discussion presented in Section 4 does not clearly convey the effectiveness of the techniques for the new theories proposed by the author, primarily because the examples discussed are well-known cases in the literature.

Report

The manuscript concerns a systematic construction of 3D mirror dual theories for Argyres-Douglas theories obtained by compactifying 6D (0,2) ADE theories. The approach is based on three steps: 1) Using the punctured sphere representation of AD theories, 2) Converting a puncture into a 3D N=4 quiver tail, and 3) Gluing the different quiver parts together.

For A and D types, the 3D mirrors are quiver gauge theories, while for E types, the mirrors are argued to be constructed by gauging strongly coupled 3D SCFTs.

Requested changes

1- The entire paper requires a major revision of punctuation, starting with the abstract and listings such as "1):". Additionally, figures are referenced inconsistently (e.g., "figure. 1"). This occurs throughout the paper.

2- Numerous sentences need revision to correct grammar and typos.

3- In Eq. (2.1), the range of b is not provided.

4- In Table 2, the superscript (1) and (2) for C-algebras is not explained.

5- Eq. (2.7) is unclear. What is the index "i" summed over, and are Y's the actual Young tableaux or only their sizes? The same issue applies to other equations like (3.12) and (3.13).

6- In Figure 8, the phrase "red puncture is sprayed as U(n1)×U(n2)" needs clarification. Additionally, the origin of the red bifundamental should be explained.

7- On page 18, there are boldface m's within some in-text equations. The reason for this formatting should be clarified.

8- On page 23 and in Figure 20, the brane configuration does not appear to be correct. There should be orientifolds. In Figure 20, Configuration B) seems incorrect as it has seven D5 branes. The brane moves should reference the earlier work by Bo Feng, preceding [36]. Configuration C) in Figure 20 also needs clarification on how the change to
O(1) is explained compared to the SO type nodes.

9- In Section 4.1, the (A,A) theories should be referenced to arXiv: 1006.3435.

10- On page 32, a boldface "a" appears that is incorrect. It should be formatted as "a" in a math environment.

11- It is unclear how the discussion in Section 4.1 translates to the new examples proposed by the author.

12- On page 34, the statement "The Coulomb branch gluing is the mirror of Higgs branch gluing, see Figure 32" is not adequately supported by Figure 32.

13- On page 37, the first enumerated point states: "The T(G) theory [36]: these theories have a Higgs branch and a Coulomb branch. The flavor symmetry on the Higgs branch is G while the flavor symmetry on the Coulomb branch is G ." This seems incorrect. For example, in T(SU(n)), both the Higgs and Coulomb branch global symmetries are PSU(n), while the GNO-dual of PSU(n) is SU(n).

14- Section 3.4 requires more detailed explanation, as the exceptional case is the most unusual and potentially most interesting. It would be beneficial for the author to expand statements such as "The interested reader can work out the mirror for other cases" into more substantial content.

Recommendation

Ask for major revision

---

## Round 1 · Referee Report · Anonymous (Referee 2) · 2024-6-10

Strengths

1- The paper gives a systematic construction of 3d mirrors of S1 compactifications of a large class of 4d N=2 SCFTs, which includes a lot of new results.

Weaknesses

1- The construction of 3d mirrors, although certainly clear for the author, is not very explicitly given. Precise and specific examples would greatly help.

2- The status of the proposed construction is unclear: is it a conjecture based on examples, or is there a systematic proof / justification?

3- Little comparison is made with existing literature.

Report

The paper discussed a large class of 4d N=2 SCFTs which are constructed in the class S formalism. For each of these theories, a systematic procedure is given to obtain a 3d N=4 quiver theory which is 3d mirror to the S^1 compactification of the initial 4d theory. This is a valuable work, as it reproduces a lot of known results in a unified way, while also providing a wealth of new results.

However, the presentation is sometimes difficult to follow. The concepts of red / black / blue punctures for class S theories is abundantly used, along with further generalizations. While this is partly reviewed in the introduction and Section 2, the paper is hardly self-contained. In addition, the way one associates Young tableaux to the punctures how this is further translated in the 3d mirror is not explicit (e.g. in Figures 1, 8, 10, 13, etc). It would greatly help to have some explicit examples as early as Section 2.

In addition, most main claims are simply stated as facts, it being unclear what is their epistemological status. Can they be derived from the Hitchin system / a brane construction / field theory arguments, or are they just guesses based on computations of examples? It would also be useful and relevant to compare the results to the related works that are, in the present version of the draft, only mentioned in passing in footnote 2.

Finally, there are a lot of typos and misprints in the text and the figures.

The results are novel and important, so the paper deserves publication, but after major improvements in its presentation.

Requested changes

1- Give a precise formula for the proposed mirror pairs, e.g. in Figure 8, give the Young Tableaux corresponding to each puncture, and explain what are n1 and n2 in the class S picture. Similarly in Figure 10, Figure 12 (what are the ni in the class S picture?), Figure 13, etc.

2- Explain how the rules for gluing are derived or inferred from examples.

3- Compare the results to those in the literature, in particular refs [17-20].

4- Improve the clarity of explanations of "known concepts", e.g.

a) The precise definition and physical significance of the black / blue / red punctures (the example shown in eqs 2.4 and 2.5 is unclear: there seem to be a typo in the exponent of z, and I don't understand how (2.4) is generic, as it depends on only 2 coefficients).
b) The comments after equation (2.6) are not sufficient to determine the precise form of the Young Tableaux
c) Is it possible to generalize the construction of eqs (2.3) to more than 2 blocks?
d) What is the range of the parameter b?
e) What is the definition of the dense open set MHit and what is its relevance for the computations in the paper?
f) Footnote 4 is unclear.

5- A few more minors typos:

a) Page 2, item 2, black should be blue.
b) Remove the full stops after "figure".
c) Page 1, "which we call it", "it" should be removed.
d) Figure 1, what is ak?
e) Below (2.1), "principle" should be "principal"
f) Page 8, the reference to (3.6) should be to (2.3)
g) Young Tableau has no x when singular
h) Figure 6, Col should be Coul
i) The punctures in Figure 7 should be red, why is a different symbol used?
j) Figure 20 should contain orientifold planes

Recommendation

Ask for major revision

---

## Editorial Decision

awaiting_resubmission